# Unlearning-based Neural Interpretations

**Ching Lam Choi***
CSAIL, Department of EECS
Massachusetts Institute of Technology
`chinglam@mit.edu`

**Alexandre Duplessis**
Department of Computer Science
University of Oxford
`alexandre.duplessis@cs.ox.ac.uk`

**Serge Belongie**
Pioneer Centre for AI
University of Copenhagen
`s.belongie@di.ku.dk`

## Abstract

Gradient-based interpretations often require an anchor point of comparison to avoid saturation in computing feature importance. We show that current baselines defined using static functions—constant mapping, averaging or blurring—inject harmful colour, texture or frequency assumptions that deviate from model behaviour. This leads to accumulation of irregular gradients, resulting in attribution maps that are biased, fragile and manipulable. Departing from the static approach, we propose `UNI` to compute an (un)learnable, debiased and adaptive baseline by perturbing the input towards an *unlearning direction* of steepest ascent. Our method discovers reliable baselines and succeeds in erasing salient features, which in turn locally smooths the high-curvature decision boundaries. Our analyses point to unlearning as a promising avenue for generating faithful, efficient and robust interpretations.

## 1 Introduction

The utility of large models is hampered by their lack of explainability and robustness guarantees. Yet breakthroughs in language modelling (Meta, 2024; Anthropic, 2024; Jiang et al., 2023; Google, 2024; Achiam et al., 2023) and generative computer vision (Rombach et al., 2022; Liu et al., 2023; Deepmind, 2024; Brooks et al., 2024) yield promising high-stakes applications, spanning domains of healthcare, scientific discovery, law and finance. As such, being able to interpret these models has become a primary concern for researchers, policymakers and the general populace, with international calls for explainability, accountability and fairness in AI decision-making (European Commission, 2021; White House OSTP, 2022; Bengio et al., 2023). To this end, recent works focus on the 2 main directions of making models *inherently explainable* (Böhle et al., 2022; Brendel & Bethge, 2018; Koh et al., 2020; Bohle et al., 2021; Chen et al., 2019; Ross et al., 2017) and *post-hoc interpretable* (Bau et al., 2017; Kim et al., 2018; Zhou et al., 2018; Ghorbani et al., 2019b). Unfortunately, the former is marred by the status quo of proprietary models and prohibitive training costs. This motivates seeking robust attributions which reliably explain model predictions, to facilitate better risk assessment and trade-off calibration (Böhle et al., 2022; Doshi-Velez & Kim, 2017).

Post-hoc methods explain a black-box model's output by attributing its decision back to predictive features of the input. They achieve this via leveraging components of the model itself (*e.g.* gradients and activations), or through approximation with a simpler, interpretable simulator. A desirable post-hoc explanation should exhibit *high faithfulness* – to be rationale-consistent (Yeh et al., 2019; Atanasova et al., 2020) with respect to a model's decision function; *low sensitivity* – to yield reliably similar saliency predictions for input features in the same local neighbourhood (Alvarez Melis & Jaakkola, 2018; Ghorbani et al., 2019b); *low complexity* – the explanation should be functionally simpler and more understandable than the original black-box model (Bhatt et al., 2021).

Gradient-based saliency methods are widely used for feature attribution, due to their simplicity, efficiency and post-hoc accessibility. This can be further decomposed into 3 families: perturbative,

---

*Research work completed during an internship at the Pioneer Centre for AI and University of Copenhagen.

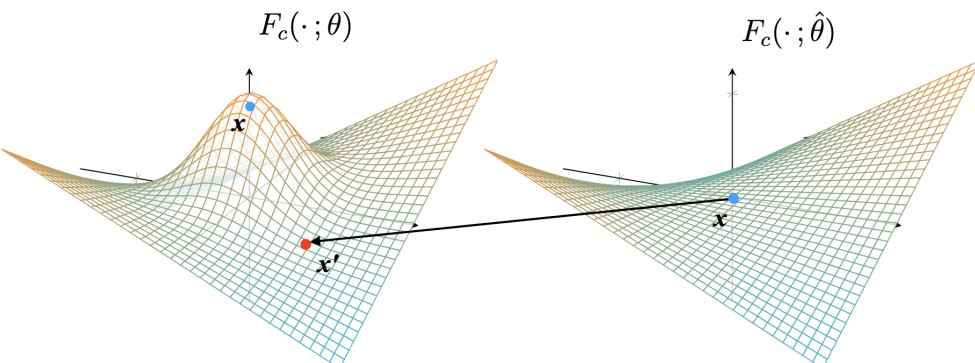

Figure 1: *Left:* Confidence of original model $\theta$ at image $x$ and baseline $x'$. *Right:* Confidence of *unlearned* model $\hat{\theta}$ at image $x$. After unlearning in the model space $\theta \longmapsto \hat{\theta}$, we optimise the baseline to match the unlearned input confidence, such that $F_c(x'; \theta) \approx F_c(x; \hat{\theta})$.

backpropagative and path-based, which we detail in Section 6. Gradient-based attribution is intuitive since the first-order derivative reveals which features significantly influence the model's classification decision. However, naively using local gradients yields unfaithful attributions due to saturation, where the non-linear output function flattens in vicinity of the input and zero gradients are computed (Sundararajan et al., 2017; 2016). To improve gradient-sensitivity, later methods introduce a baseline input for reference, and backpropagate the difference in activation scores on a path between the reference and image-of-interest (Shrikumar et al., 2016; Sundararajan et al., 2017). The baseline is chosen to be devoid of predictive features and far away from the saturated local neighbourhood. However, such methods accumulate gradient noise when interpolating from the baseline to the input, leading to high local sensitivity (Ancona et al., 2018). Consequently, attribution maps become disconnected, sparse and irregular, where the saliency scores fluctuate wildly between neighbouring pixels of the same object and are visually noisy (Adebayo et al., 2018). This noise accumulation has two root causes—a *poorly chosen baseline* and *high-curvature output manifold* along the path features. Previous works (Sturmfels et al., 2020; Xu et al., 2020) have sought better baselines by empirically comparing between using a black image, a gaussian noised image, a gaussian blurred image, a uniformly noised image, an inverted colour image, as well as averaging attributions over several baseline choices. However, the correct baseline to represent a lack of salient features depends heavily on the specific classification task, on the trained model and on the input image. Indeed, the optimal baseline varies for each task–model–image combination (Akhtar & Jalwana, 2023); the baseline problem remains largely unsolved. Turning to the second problem of high-curvature output manifold, because trained neural networks exhibit approximately piece-wise linear decision boundaries (Goodfellow et al., 2014), inputs near function transitions are vulnerable to perturbative attacks. By simply adding norm-bounded, imperceptible adversarial noise to the input image, attackers can dramatically alter the attribution map without changing the model's class prediction (Ghorbani et al., 2019a; Dombrowski et al., 2019). Methods of mitigation include explicit smoothing via averaging over multiple noised gradient attributions (Smilkov et al., 2017); adaptively optimising the integration path of attribution (Kapishnikov et al., 2021); imposing an attribution prior during training and optimising it at each step (Erion et al., 2021). However, all of these proposals starkly increase the complexity of attribution, requiring computationally costly forward and backward propagation steps.

To tackle the problematic triad of *1. post-hoc attribution biases, 2. poor baseline definition, 3. high-curvature output manifold*, we propose `UNI` to discover debiased baselines by locally *unlearning* inputs, *i.e.* perturbing them in the unlearning direction of steepest ascent, as visualised in Figure 1. Towards better baselines, our unlearned reference is by definition explicitly optimised to lower output class confidence and can empirically erase or occlude salient features. We also say that the unlearned baseline is specific and featureless w.r.t. each task–model–input combination. Unlike the practice of using a black image baseline—which creates a post-hoc colour bias that darker pixels are less likely to be salient, `UNI` does not impose additional, pixel-wise colour, scale or geometric assumptions that are not already present in the model itself. Finally, we address the high-curvature decision boundaries problem by realising that this is a product of the training process—targeted unlearning smooths the decision boundary of the model within the vicinity of the input. For a more detailed overview on the principle of machine unlearning, we refer the reader to Section 6 of the supplement. We empirically

verify this local smoothing effect by measuring the normal curvature of the model function before and after unlearning; we also demonstrate that unlearning makes attributions resistant to perturbative attacks. Our contributions can be summarised as follows:

1. *Post-hoc attribution can impose new biases.* We approach the baseline challenge from the fresh lens of post-hoc biases. We show that static baselines (*e.g.* black, blurred, random noise) inject additional colour, texture and frequency assumptions that are not present in the original model's decision rule, which leads to explanation infidelity and inconsistency.

2. *A well-chosen baseline is specific and featureless.* We establish theoretically grounded principles for sound baseline definitions, by formalising the idea of an "absence of signal" through an unlearning direction of steepest ascent in model loss. By unlearning predictive features in the model space and matching this reference model's activations with a perturbation in the input space, we introduce a new definition of "feature absence" and a novel attribution algorithm.

3. *Unlearning reduces the curvature of decision boundaries and increases robustness.* Targeted unlearning simulates function statistics of unseen data, and smooths the curvature of the output manifold around the sample. This is characterised by low geodesic path curvature and bounded principal curvature of the output surface. This points to reduced variability of the gradient vector under small-norm input perturbations, leading to better attribution robustness and faithfulness.

## 2 PRELIMINARIES

We consider feature attribution for trained deep neural networks within image classification. Informally, we seek to assign scores to each pixel of an image for quantifying the pixel's influence (sign and magnitude) on the predicted output class confidence. It is noteworthy that attributions can be signed: a negative value indicates that removing the pixel increases the target class probability.

### 2.1 NOTATION

The input (feature) space is denoted as $\mathcal{X} \subset \mathbb{R}^{d_X}$, where $d_X$ is the number of pixels in an image. The output (label) space is $\mathcal{Y} \subset \mathbb{R}^{d_Y}$; $\mathcal{Y}$ is the set of all probability distributions on the set of classes. The model space is denoted as $\mathcal{F} \subset \mathcal{Y}^{\mathcal{X}}$. A trained model $F : x \mapsto (F_1(x), ..., F_{d_Y}(x))$ returns the probability score $F_c(x)$ of each class $c$. Attribution methods are thus functions $\mathcal{A} : \{1, ..., d_X\} \times \mathcal{F} \times \{1, ..., d_Y\} \times \mathcal{X} \to \mathbb{R}$, where $\mathcal{A}(i, F, c, x)$ is the importance score of pixel $i$ of image $x$ for the prediction made by $F_c$. For convenience, we use the shorthand $\mathcal{A}_i(x)$ to refer to the attributed saliency score of a pixel $i$ for a specific class prediction $c \in \{1, ..., d_Y\}$. We express a linear path feature as $\gamma(x', x, \alpha) : \mathbb{R}^{d_X} \times \mathbb{R}^{d_X} \times [0, 1] \to \mathbb{R}^{d_X}$, where $\gamma = (1 - \alpha)x' + \alpha x$ and employ shorthands $\gamma(0) = x', \gamma(1) = x$.

## 3 GRADIENT-BASED ATTRIBUTIONS IN A NUTSHELL

### 3.1 LIMITATIONS

Taking the local gradients of a model's output confidence map $F_c(x)$ – for target class $c$ – is a tried and tested method for generating explanations. Commonly termed Simple Gradients (Erhan et al., 2009; Baehrens et al., 2010; Simonyan et al., 2013), $\mathcal{A}_i^{\text{SG}}(x) = \nabla_{x_i} F_c(x)$ can be efficiently computed for most model architectures. However, it encounters output saturation when activation functions like ReLU and Sigmoid are used, leading to zero gradients (hence null attribution) even for important features (Sundararajan et al., 2017; 2016). DeepLIFT (Shrikumar et al., 2016) reduces saturation by introducing a "reference state". A feature's saliency score is decomposed into positive and negative contributions by backpropagating and comparing each neuron's activations to that of the baseline. Integrated Gradients (IG) (Sundararajan et al., 2017) similarly utilises a reference, black image and computes the integral of gradients interpolated on a straight line between the image and the baseline.

$$\mathcal{A}_i^{\text{IG}}(x) = (x_i - x_i') \int_{\alpha=0}^{1} \nabla_{x_i} F_c\left(x' + \alpha(x - x')\right) d\alpha \tag{1}$$

Practically, the integral is approximated by a Riemann sum. Of existing methods, IG promises desirable, game-theoretic properties of "sensitivity", "implementation invariance", "completeness"

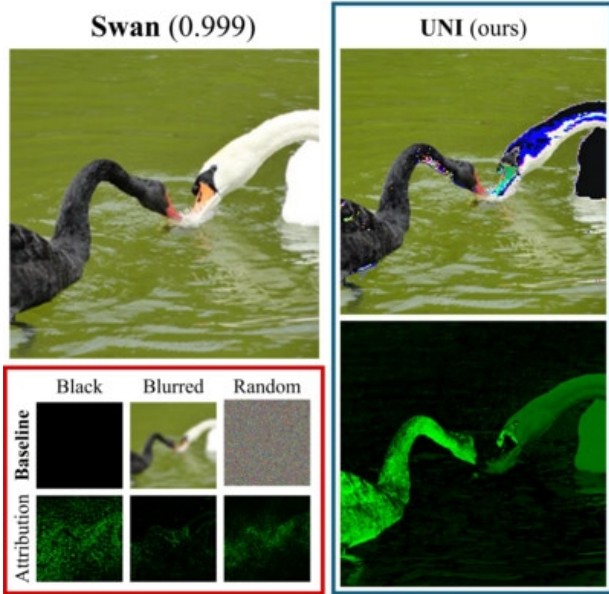

Figure 2: We visualise post-hoc biases imposed by static baselines—black baseline (colour), blurred (texture), random (frequency). `UNI` learns to mask out predictive features used by the model, generating reliable attributions.

**Algorithm 1** `UNI`: unlearning direction, baseline matching and path-attribution

1: **Given** model $F(\cdot, \theta)$; inputs $(x, y)$

2: **Choose** unlearning step-size $\eta$; PGD steps $T$, budget $\varepsilon$, step-size $\mu$; Riemann approximation steps $B$

3: **Initialise** perturbation $\delta^0$

4: **Unlearning direction.**
$$\hat{\theta} = \theta + \eta \frac{\nabla_\theta \mathcal{L}(F_c(x;\theta),y)}{\|\nabla_\theta \mathcal{L}(F_c(x;\theta),y)\|}$$

5: **for** $t = 0, \cdots, T-1$ **do**
$$\mathcal{C} = D_{KL}(F(x;\hat{\theta}) \parallel F(x+\delta^t;\theta))$$
$$\delta^{t+1} = \delta^t - \mu \nabla_\delta \mathcal{C}$$
$$\delta^{t+1} = \varepsilon \frac{\delta^{t+1}}{\|\delta^{t+1}\|}$$

6: **end for**

7: **Baseline** definition $x' = x + \delta^T$

8: **Attributions** computation: $\mathcal{A}_i^{\text{UNI}}(x)$
$$= \frac{(x_i - x_i')}{B} \sum_{k=1}^{B} \nabla_{x_i} F_c \left( x' + \frac{k}{B}(x - x'); \theta \right)$$

and "linearity". We consequently focus on analysing and developing the IG framework, though the proposal to unlearn baselines can be applied to most mainstream gradient-based saliency methods. Despite the advantages of IG, its soundness depends on a good *baseline definition*—an input which represents the "absence" of predictive features; also on having stable *path features*—a straight-line of increasing output confidence along the path integral from baseline to target image. In the conventional setting where a black image is used, Akhtar & Jalwana (2023) prove that IG assumptions are violated due to ambiguous path features, where extrema of model confidences lie along the integration path instead of at the endpoints of the baseline (supposed minimum) and input image (supposed maximum). Sturmfels et al. (2020) enumerate problems with other baselines obtained via gaussian blurring, maximum distance projection, uniform noise. Despite the diversity of baseline alternatives, no candidate is optimal for each and every attributions setting. For instance, models trained with image augmentations (*e.g.* colour jittering, rescaling, gaussian blur) yield equivalent or even higher confidences for blurred and lightly-noised baselines—we need baselines that are well-optimised for each task–model–input combination. Without principled baselines, problems of non-conformant intermediate paths and counter-intuitive attribution scores will doubtlessly persist.

### 3.2 POST-HOC BIASES ARE IMPOSED

Since the baseline represents an absence of or reduction in salient features, static baseline functions (*e.g.* black, blurred, noised) implicitly assume that similar features (*e.g.* dark, smooth, high-frequency) are irrelevant for model prediction. To illustrate this intuition, we can consider IG with a black baseline, wherein it becomes more difficult to attribute dark but salient pixels. Due to the colour bias that "near-black features are unimportant", the term $(x_i - x_i')$ is small and requires a disproportionately large gradient $\nabla_{x_i} F_c(\cdot)$ to yield non-negligible attribution scores. Indeed, this is what we observe in Figures 2, 3, 11, where darker features belonging to the object-of-interest cannot be reliably identified. We further empirically verify that each static baseline imposes its own post-hoc bias by experimenting on ImageNet-C (Hendrycks & Dietterich, 2019). Corresponding to the 3 popular baseline choices for IG (all-black, gaussian blurred, gaussian noised), we focus on the families of *digital* (brightening and saturation), *blur* (gaussian and defocus blur) and *noise* (gaussian and shot noise) common-corruptions. Figures 4, 12 demonstrate that IG with a blurred baseline fails to attribute blurred inputs due to saturation and overly smoothed image textures; Figures 5, 13 visualise how a noised IG baseline encounters high-frequency noise and outputs irregular, high-variance attribution scores, even for

adjacent pixels belonging to the same object. We crucially emphasise that such colour, texture and frequency biases are not present naturally in the pre-trained model but rather injected implicitly by a suboptimal choice of static baseline. The observation that poor baseline choices create *attribution bias* has so far been overlooked. As such, we depart entirely from the line of work on alternative static baseline towards adaptively (un)learning baselines with gradient-based optimisation. *UNI eliminates all external assumptions except for the model's own predictive bias.*

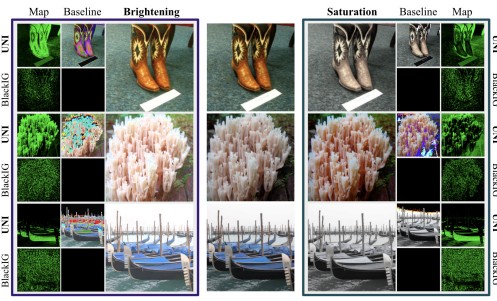

Figure 3: When the brightness or saturation is altered, IG with a black baseline fails to identify dark features, such as the boat's hull (R3) or the top of the boot (R1).

Table 1: **Path monotonicity scores** with Spearman correlation coefficient (higher = better). Integrating from a "featureless" baseline to the sample should give a path of monotonically increasing prediction confidence.

|  | UNI | IG | BlurIG | GIG |
|---|---|---|---|---|
| ResNet-18 | **.97** ±.222 | .69 ±.460 | .57 ±.576 | .45 ±.476 |
| Eff-v2-s | **.95** ±.258 | .28 ±.615 | .34 ±.613 | .38 ±.437 |
| ConvNeXt-T | **.99** ±.121 | .76 ±.379 | .77 ±.486 | .46 ±.485 |
| VGG-16-bn | **.94** ±.286 | .69 ±.474 | .60 ±.544 | .46 ±.479 |
| ViT-B-16 | **.89** ±.396 | .71 ±.399 | .27 ±.648 | .44 ±.468 |
| SwinT | **.97** ±.189 | .88 ±.326 | .88 ±.482 | .45 ±.474 |

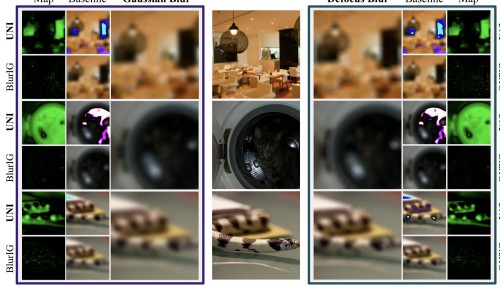

Figure 4: Under gaussian or defocus blur, IG with a blurred baseline suffers from saturation; has overly smooth texture; does not yield meaningful features.

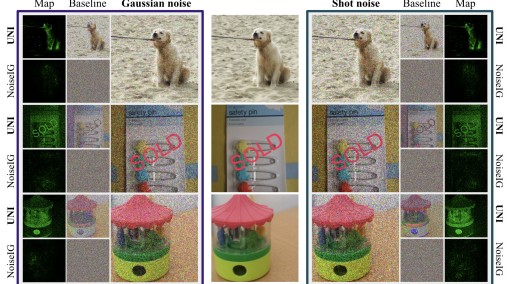

Figure 5: Gaussian and shot noise create visual artifacts prominent in noised-baseline IG. Frequency bias leads to disparate scores for adjacent pixels.

## 4 UNI: UNLEARNING-BASED NEURAL INTERPRETATIONS

### 4.1 BASELINE DESIDERATA

A desirable baseline should preserve the game-theoretic properties of path-attribution (Section 3.1) and refrain from imposing post-hoc attribution biases (Section 3.2). For every given task-model-image triad, a well-chosen baseline should be *1. image-specific*—be connected via a path feature of low curvature to the original image; *2. reflect only the model's predictive biases*—salient image features should be excluded from the baseline; be *3. less task-informative than the original image*—interpolating from the baseline towards the input image should yield a path of increasing predictive confidence. We now introduce the UNI pipeline: first, unlearn predictive information in the model space; then, use activation-matching between unlearned and trained models to mine a featureless baseline in the image space; finally, interpolate along the low-curvature, conformant and consistent path from baseline to image to compute reliable explanations in the attributions space. Figure 8 visuals and Table 7 results attest to UNI's ability to compute specific, unlearned baselines for attribution.

### 4.2 DESIRABLE PATH FEATURES

**Proximity** The meaningfulness of the attributions highly depends on the meaningfulness of the path. We aim for a smooth transition between absence and presence of features; and this intuitively cannot be achieved if the baseline and input are too far apart. Srinivas & Fleuret (2019) formalises this intuition through the concept of *weak dependence*, and proves that this property can only be

compatible with completeness in the case where the baseline and the input lie in the same connected component (in the case of piecewise-linear models). An obvious implementation of this proximity condition in the general case is to bound the distance $||x - x'||$ to a certain value $\varepsilon$. This is strictly enforced in Algorithm 1 by normalising the perturbation at each step $t$.

**Low Curvature.** The curvature of the model prediction along the integrated path has been identified Dombrowski et al. (2019) as one of the key factors influencing both the sensitivity and faithfulness of the computed attributions. We substantiate the intuition that a smooth and regular path is preferred by analysing the Riemannian sum calculation. Assuming that the function $g : \alpha \in [0, 1] \mapsto \nabla F_c (x' + \alpha(x - x'))$ is derivable with a continuous derivative (i.e. $\mathcal{C}^1$) on the segment $[x', x]$, elementary calculations and the application of the Taylor-Lagrange inequality give the following error in the Riemann approximation of the attribution,

$$\left| (x_i - x_i') \int_{\alpha=0}^{1} g(\alpha)\, d\alpha - \frac{(x_i - x_i')}{B} \sum_{k=1}^{B} g\left(\frac{k}{B}\right) \right| \leq \frac{M||x - x'||^2}{2B} \tag{2}$$

where $M = \max_{\alpha \in [0,1]} \frac{\mathrm{d}g}{\mathrm{d}\alpha} = \max_{\alpha \in [0,1]} \frac{\partial^2 F_c(x' + \alpha(x - x'))}{\partial \alpha^2}$ exists by continuity of $g'$ on $[0, 1]$. Thus, lower curvature along the path implies a lower value of the constant $M$, which in turn implies a lower error in the integration calculation. A smaller value $B$ of Riemann steps is needed to achieve the same precision. More generally, a low curvature (i.e. eigenvalues of the hessian) on and in a neighbourhood of the baseline and path reduces the variability of the calculated gradients under small-norm perturbations, increasing the sensitivity and consistency of the method. Empirically, we observe a much lower curvature of paths computed by `UNI`, as per Table 1 and Appendix Figures 20, 21, 22, 23, 24, 25. Figure 10 also confirms the increased robustness to Riemann sum error induced.

**Monotonic.** Intuitively, the path $\gamma$ defined by interpolating from the "featureless" baseline $x'$ to the input image $x$ should be *monotonically increasing* in output class confidence. At the image level, for all $j, k$ such that $j \leq k$, since $||\gamma(j) - x|| \geq ||\gamma(k) - x||$, therefore the predictive confidence should be non-decreasing and order-preserving: $F_c(\gamma(j)) \leq F_c(\gamma(k))$. Constraining $\gamma$ to be monotonically increasing suffices to satisfy a weak version of the criteria for *valid path features* (Akhtar & Jalwana, 2023): $\mathrm{sgn}(\nabla_x F_c(x)) \cdot \mathrm{sgn}(\nabla_{\tilde{x}} F_c(x')) = 1$ is naturally met.

### 4.3 Effects of Unlearning and Matching

We explain the success of `UNI` with the illustrative example of a three gaussians mixture model. Figure 6 computes unlearning and activation matching for a model learned on three data points with gradient descent. $F$ is chosen to be the output of the three gaussian components $(G_1, G_2, G_3)$. Note that the perturbation is not $\varepsilon$-normalised for clearer visualisation. We highlight two observations:

- The `UNI` path is monotonous, of low-curvature and proximal. Conversely, the path to the random baseline is long, non-monotonous, and goes through several zones of high second derivative.
- Optimizing KL divergence on $(G_1, G_2, G_3)$ produces a better baseline. Figure 6b visualises the unlearning objective (*i.e.* the target probability after unlearning), which gives four points of intersection with the base model ($a, b, c$ and $d$). By constraining proximity of the baseline with the $\varepsilon$ parameter, we restrain the optima found by gradient descent (on the global probability) to the closest two points $a$ and $b$. `UNI` is then able yield the more optimal of the two, by optimising on each gaussian output. In fact, the idea of activation matching is to satisfy the crucial weak dependence property for conformal path attribution (Akhtar & Jalwana, 2023). Since modern ReLU networks have decision boundaries representable as piecewise linear functions (Xiong et al., 2020), activation matching supervises the baseline to use the same (piecewise linear) weights. In our case, we want to find a baseline for which $G_1$ and $G_2$ do not play a role, which is not the case for $a$. This is why Algorithm 1 optimises on $F$ and not on $F_c$.

Finally, $\varepsilon$ normalisation serves to regularise baseline GD learning and account for pathological cases where the locally shortest path would lead to further intersections than the closest one.

## 5 Experiments

We experiment on ImageNet-1K (Deng et al., 2009), ImageNet-C (Hendrycks & Dietterich, 2019) and compare against various path-based and gradient-based attribution methods. This includes IG

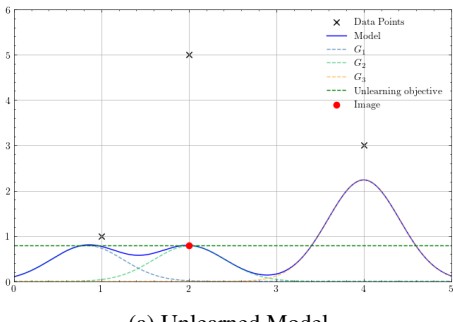 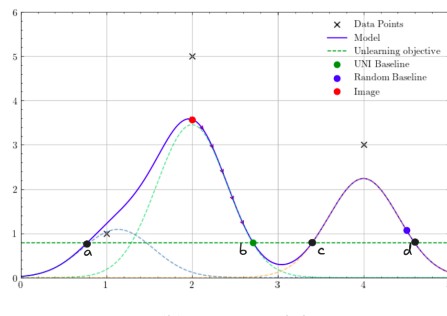

| (a) Unlearned Model | (b) Base Model |

Figure 6: `UNI` baseline on a Gaussian mixture model of three gaussians $G_1, G_2, G_3$, each of fixed variance, parametrised by their mean and a scaling factor. (b) shows the model trained on the three datapoints (1,1), (2, 5) and (4, 3), while (a) shows the model after one gradient ascent step on the datapoint (2, 5). The path between `UNI` Baseline and the image is highlighted by arrows in (b).

(Sundararajan et al., 2017), BlurIG (Xu et al., 2020), GIG (Kapishnikov et al., 2021), AGI (Pan et al., 2021), GBP (Springenberg et al., 2014) and DeepLIFT (Shrikumar et al., 2016). We consider a diverse set of pre-trained computer vision backbone models (Paszke et al., 2019), including ResNet-18 (He et al., 2016), EfficientNet-v2-small (Tan & Le, 2021), ConvNeXt-Tiny (Liu et al., 2022), VGG-16-bn (Simonyan & Zisserman, 2015), ViT-B_16 (Dosovitskiy et al., 2020) and Swin-Transformer-Tiny (Liu et al., 2021). Unless otherwise specified, we the following hyperparameters: unlearning step size $\eta = 1$; $l_2$ PGD with $T = 10$ steps, a budget of $\varepsilon = 0.25$, step size $\mu = 0.1$; Riemann approximation with $B = 15$ steps. We further extend `UNI` to the NLP domain, to interpret generative language models using activation patching (Heimersheim & Nanda, 2024). `UNI` complements activation matching by computing a stable baseline without trading off attribution scalability, as observed in Appendix Table 8 and Figure 9. Our results verify `UNI`'s high faithfulness, stability and robustness.

Table 2: *MuFidelity scores* measure the correlation between a subset of pixels' impact on the output (*i.e.* change in predictive confidence) and assigned saliency scores. Since attribution methods can yield strong positive or negative correlations, we report the absolute scores.

|  | UNI | IG | BlurIG | GIG | AGI | GBP | DeepLIFT |
|---|---|---|---|---|---|---|---|
| ResNet-18 | **.12** $\pm.124$ | .06 $\pm.068$ | .07 $\pm.076$ | .07 $\pm.080$ | .10 $\pm.110$ | .09 $\pm.094$ | .08 $\pm.082$ |
| EfficientNetv2s | **.06** $\pm.046$ | .05 $\pm.043$ | .05 $\pm.044$ | .05 $\pm.044$ | .06 $\pm.045$ | .05 $\pm.043$ | .05 $\pm.043$ |
| ConvNeXt-Tiny | .16 $\pm.115$ | .11 $\pm.086$ | .15 $\pm.121$ | **.18** $\pm.149$ | .17 $\pm.131$ | .09 $\pm.072$ | .11 $\pm.084$ |
| VGG-16-bn | **.18** $\pm.141$ | .08 $\pm.066$ | .09 $\pm.076$ | .13 $\pm.108$ | .14 $\pm.104$ | .13 $\pm.108$ | .10 $\pm.082$ |
| ViT-B_16 | **.15** $\pm.114$ | .10 $\pm.074$ | .10 $\pm.077$ | .11 $\pm.079$ | .14 $\pm.104$ | .09 $\pm.070$ | .10 $\pm.072$ |
| Swin-T-Tiny | **.13** $\pm.100$ | .09 $\pm.071$ | .12 $\pm.102$ | .12 $\pm.104$ | .13 $\pm.102$ | .09 $\pm.069$ | .10 $\pm.076$ |

## 5.1 FAITHFULNESS

We report MuFidelity scores (Bhatt et al., 2021), *i.e.* the faithfulness of an attribution function $\mathcal{A}$, to a model $F$, at a sample $x$, for a subset of features of size $|S|$, given by $\mu_f(F, \mathcal{A}; x) = \text{corr}_{S \in \binom{[d]}{|S|}} \left( \sum_{i \in S} \mathcal{A}(i, F, c, x),\ F_c(x) - F_c(x_{[x_s = \bar{x}_s]}) \right)$. We record the (absolute) correlation coefficient between a randomly sampled subset of pixels and their attribution scores. In line with open source exemplars (Fel et al., 2022a), we set $|S|$ to be 25% of the total pixel count (slightly higher than the referenced 20%) as is required to adjust for ImageNet's complexity and for obtaining less noisy measurements across all baseline methods. As from Table 2, `UNI` outperforms other methods across all settings but one, indicating high faithfulness. We supplement these numbers with visual comparisons in Appendix Figures 14, 15, 16, 17, 18, 19 against IG (black and noised baselines), BlurIG, GIG, AGI, GBP, DeepLift. Furthermore, we report deletion and insertion scores (Petsiuk et al., 2018)—a causally-motivated evaluation metric for interpretability methods—which measures the decrease (deletion) or increase (insertion) of a model's output confidence as salient pixels are removed (from the original image) or inserted (into a featureless baseline). A steep drop in model confidence under pixel deletion results in a desirable and small area under the curve (AUC) score; a sharp rise under pixel insertion results in a desirably large AUC. Salient pixels are removed in descending order of importance, as identified by the tested interpretability method. We evaluate with

a step size of 10% and average over 10,000 random image samples, where at each step, the next-10% most salient pixels are removed or inserted for inference. `UNI` reliably identifies pixels which are crucial for sample classification, achieving marked improvements especially in insertion AUC scores.

Table 3: *Deletion AUC ↓ measures how confidence drops as pixels are removed (lower = better).*

|  | UNI | IG | BlurIG | GIG | AGI | GBP | DeepLIFT |
|---|---|---|---|---|---|---|---|
| ResNet-18 | **.06** ±.128 | .10 ±.174 | .27 ±.252 | .11 ±.150 | .13 ±.147 | .08 ±.160 | .13 ±.165 |
| EfficientNetv2s | .19 ±.212 | .26 ±.217 | .50 ±.158 | .19 ±.216 | **.18** ±.207 | .23 ±.163 | .27 ±.215 |
| ConvNeXt-Tiny | **.11** ±.139 | .16 ±.164 | .46 ±.172 | .21 ±.160 | .17 ±.123 | .16 ±.099 | .21 ±.162 |
| VGG-16-bn | **.08** ±.143 | .12 ±.181 | .18 ±.241 | .10 ±.163 | .14 ±.178 | .14 ±.194 | .12 ±.186 |
| ViT-B_16 | .14 ±.185 | .22 ±.207 | .60 ±.166 | .17 ±.190 | **.13** ±.152 | .23 ±.141 | .17 ±.189 |
| Swin-T-Tiny | **.13** ±.181 | .22 ±.217 | .47 ±.174 | .22 ±.207 | .21 ±.172 | .21 ±.123 | .23 ±.207 |

Table 4: *Insertion AUC ↑ measures how confidence rises as pixels are inserted (higher = better).*

|  | UNI | IG | BlurIG | GIG | AGI | GBP | DeepLIFT |
|---|---|---|---|---|---|---|---|
| ResNet-18 | **.64** ±.138 | .26 ±.045 | .34 ±.131 | .36 ±.048 | .56 ±.068 | .11 ±.066 | .18 ±.042 |
| EfficientNetv2s | **.64** ±.227 | .38 ±.127 | .51 ±.283 | .37 ±.138 | .38 ±.204 | .23 ±.192 | .37 ±.137 |
| ConvNeXt-Tiny | **.63** ±.231 | .21 ±.114 | .40 ±.252 | .56 ±.122 | .52 ±.088 | .22 ±.160 | .17 ±.162 |
| VGG-16-bn | **.56** ±.335 | .37 ±.061 | .31 ±.274 | .38 ±.071 | .47 ±.078 | .26 ±.057 | .17 ±.056 |
| ViT-B_16 | **.71** ±.237 | .32 ±.107 | .59 ±.292 | .28 ±.125 | .43 ±.089 | .35 ±.172 | .28 ±.123 |
| Swin-T-Tiny | **.68** ±.245 | .28 ±.145 | .63 ±.282 | .26 ±.153 | .25 ±.156 | .31 ±.202 | .26 ±.152 |

## 5.2 ROBUSTNESS

Next, we evaluate `UNI`'s robustness to fragility adversarial attacks on model interpretations. Following Ghorbani et al. (2019a), we design norm-bounded attacks to maximise the disagreement in attributions whilst constraining that the prediction label remains unchanged. We consider a standard $l_\infty$ attack designed with FGSM (Goodfellow et al., 2014), with perturbation budget $\varepsilon_f = 8/255$.

$$\delta_f^* = \arg \max_{\|\delta_f\|_p \le \varepsilon_f} \frac{1}{d_X} \sum_{i=1}^{d_X} d\left(\mathcal{A}(i, F, c, x), \mathcal{A}(i, F, c, x + \delta_f)\right) \tag{3}$$

$$\text{subject to} \quad \arg \max_{c'} F_{c'}(x) = \arg \max_c F_{c'}(x + \delta_f) = c$$

We report robustness results using 2 distance measures—Spearman correlation coefficient in Table 5 and top-$k$ pixel intersection score in Table 6—pre and post attack. While other methods like DeepLIFT (DL), BlurIG, Integrated Gradients (IG) are misled to output irrelevant feature saliencies, `UNI` robustly maintains attribution consistency and achieves the lowest attack attribution disagreement scores (before and after FGSM attacks) for both metrics.

## 5.3 STABILITY

We compare `UNI` and other methods' sensitivity to Riemann approximation noise, which manifests in visual artefacts and misattribution of salient features. As seen from Figures 7, 10, `UNI` reliably finds unlearned, "featureless" baselines for consistent attribution, regardless of the number of approximation steps $B \in \{1, 15, 30\}$. This is due to the low geodesic curvature of $\gamma^{\text{UNI}}$, which approximately minimises the local distance between points used in Riemann approximation.

Table 5: *Robustness*: Spearman's correlation coefficient. Higher scores indicate better path consistency pre/post FGSM attacks.

|  | UNI | IG | BlurIG | SG | DeepL |
|---|---|---|---|---|---|
| ResNet-18 | **.271** | .088 | .084 | .014 | .139 |
| Eff-v2-s | **.302** | .009 | .076 | .008 | .018 |
| ConvNeXt-T | **.292** | .010 | .127 | .011 | .012 |
| VGG-16-bn | **.290** | .143 | .098 | .014 | .108 |
| ViT-B-16 | **.319** | .018 | .066 | .023 | .023 |
| SwinT | **.271** | .088 | .084 | .014 | .139 |

Table 6: *Robustness*: Top-1000 pixel intersection. Higher percentages indicate better attribution reliability pre/post FGSM attacks.

|  | UNI | IG | BlurIG | SG | DeepL |
|---|---|---|---|---|---|
| ResNet-18 | **37.3** | 20.0 | 25.3 | 18.2 | 24.8 |
| Eff-v2-s | **39.4** | 17.4 | 23.3 | 18.6 | 18.0 |
| ConvNeXt-T | **34.8** | 15.0 | 26.2 | 16.7 | 15.1 |
| VGG-16-bn | **35.7** | 25.5 | 25.3 | 18.8 | 25.2 |
| ViT-B-16 | **40.7** | 17.1 | 21.7 | 19.6 | 17.2 |
| SwinT | **37.3** | 20.0 | 25.3 | 18.2 | 24.8 |

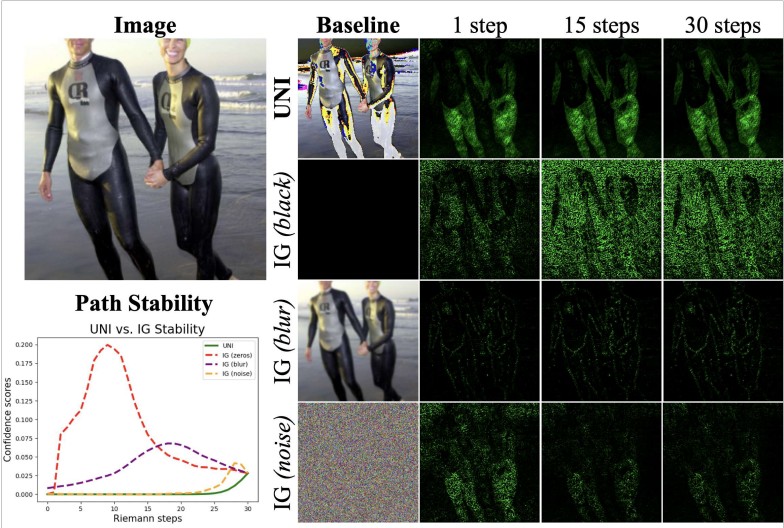

Figure 7: `UNI` path features monotonically increase in output confidence when interpolating from baseline to input. This eliminates instability and inconsistency problems caused by extrema and turning points along the Riemann approximation path, which is present in other methods.

## 6 RELATED WORK

**Machine unlearning.** We draw inspiration from the high-level principle of unlearning, which concerns the targeted "forgetting" of a data-point for a trained model, by localising relevant information stored in network weights and introducing updates or perturbations (Bourtoule et al., 2021). Formally, machine unlearning can be divided into exact and approximate unlearning (Nguyen et al., 2022). Exact unlearning seeks indistinguishability guarantees for output and weight distributions, between a model not trained on a sample and one that has unlearned said sample (Ginart et al., 2019; Thudi et al., 2022; Brophy & Lowd, 2021). However, provable exact unlearning is only achieved under full re-training, which can be computationally infeasible. Hence, approximate unlearning was proposed stemming from $\epsilon$-differential privacy (Dwork, 2011) and certified removal mechanisms (Guo et al., 2020; Golatkar et al., 2020). The former guarantees unlearning for $\epsilon = 0$, *i.e.* the sample has null influence on the decision function; the latter unlearns with first/second order gradient updates, achieving max-divergence bounds for single unlearning samples. Unlearning naturally lends itself to path-based attribution, to localise then delete information in the weight space, for the purposes of defining an "unlearned" activation. This "unlearned" activation can be used to match the corresponding, "featureless" input, where salient features have been deleted during the unlearning process. While the connection to interpretability is new, a few recent works intriguingly connect machine unlearning to the task of debiasing classification models during training and evaluation (Chen et al., 2024; Kim et al., 2019; Bevan & Atapour-Abarghouei, 2022).

**Perturbative methods.** Perturbative methods perturb inputs to change and explain outputs (Sculley et al., 2015), including LIME (Ribeiro et al., 2016), SHAP, KernelSHAP and GradientSHAP (Lundberg et al.), RKHS-SHAP (Chau et al., 2022), ConceptSHAP (Yeh et al., 2020), InterSHAP (Janzing et al., 2020), and DiCE (Kommiya Mothilal et al., 2021). LIME variants optimise a simulator of minimal functional complexity able to match the black-box model's local behaviour for a given input-label pair. SHAP (Lundberg et al.) consolidates LIME, DeepLIFT (Shrikumar et al., 2016), Layerwise Relevance Propagation (LRP) (Montavon et al., 2019) under the general, game-theoretic framework of additive feature attribution methods. For this framework, they outline the desired properties of local accuracy, missingness, consistency; they propose SHAP values as a feature importance measure which satisfies these properties under mild assumptions to generate model-agnostic explanations. However, such methods fail to give a global insight of the model's decision function and are highly unstable due to the reliance on local perturbations (Fel et al., 2022b). Bordt et al. (2022) show that this leads to variability, inconsistency and unreliability in generated explanations, where different methods give incongruent explanations which cannot be acted on. Recent works have made considerable progress, including RISE (Petsiuk et al., 2018), which strives

to causally explain model predictions by approximating the necessary saliency of pixels through random masking; Sobol (Fel et al., 2021), which adapts Sobol indices for perturbation masks towards variance-based sensitivity analysis; and FORGrad (Muzellec et al., 2023), which filters out high-frequency gradient noise induced by white-box methods (and network pooling or striding operations) and which can be complementarily applied to further `UNI`'s explanation faithfulness and efficiency.

**Backpropagative methods.**   Beginning with simple gradients (Erhan et al., 2009; Simonyan et al., 2013), this family of methods—also, LRP (Montavon et al., 2019), DeepLIFT (Shrikumar et al., 2016), DeConvNet (Zeiler & Fergus, 2014), Guided Backpropagation (Springenberg et al., 2014) and GradCAM (Selvaraju et al., 2017)—leverages gradients of the output $w.r.t.$ the input to proportionally project predictions back to the input space, for some given neuron activity of interest. Gradients of neural networks are, however, highly noisy and locally sensitive – they can only crudely localise salient feature regions. While this issue is partially remedied by SmoothGrad (Smilkov et al., 2017), we still observe that gradient-based saliency methods have higher sample complexity for generalisation than normal supervised training (Choi & Farnia, 2024) and often yield inconsistent attributions for unseen images at test time.

**Path-based attribution.**   This family of post-hoc attributions is attractive due to its grounding in cooperative game-theory (Friedman, 2004). It comprises Integrated Gradients (Sundararajan et al., 2017), Adversarial Gradient Integration (Pan et al., 2021), Expected Gradients (EG) (Erion et al., 2021), Guided Integrated Gradients (GIG) (Kapishnikov et al., 2021) and BlurIG (Xu et al., 2020). Path attribution typically relies on a baseline – a "vanilla" image devoid of features; a path—an often linear path from the featureless baseline to the target image—along which the path integral is computed for every pixel. Granular control over the attribution process comes with difficulties of defining an unambiguously featureless baseline (for each (model, image) pair) (Sturmfels et al., 2020) and then defining a reliable path of increasing label confidence without intermediate inflection points (Akhtar & Jalwana, 2023). To measure the discriminativeness of features identified by attribution methods and the extent to which model predictions depend on them, experimental benchmarks and metrics such as ROAR (Hooker et al., 2019), DiffRAOR (Shah et al., 2021), deletion/insertion score (Petsiuk et al., 2018), the Hilbert-Schmidt independence criterion (HSIC) (Novello et al., 2022) and the Pointing Game (Zhang et al., 2018a) have been proposed.

## 7 CONCLUSION

In this work, we formally discuss the limitations of current path-attribution frameworks, outline a new principle for optimising baseline and path features, as well as introduce the `UNI` algorithm for unlearning-based neural interpretations. We empirically show that present reliance on static baselines imposes undesirable post-hoc biases which are alien to the model's decision function. We account for and mitigate various infidelity, inconsistency and instability issues in path-attribution by defining principled baselines and conformant path features. `UNI` leverages insights from unlearning to eliminate task-salient features and mimic baseline activations in the "absence of signal". It discovers low-curvature, stable paths with monotonically increasing output confidence, which preserves the completeness axiom necessary for path attribution. We visually, numerically and formally establish the utility of `UNI` as a means to compute robust, meaningful and debiased image attributions.

The contributions of `UNI` extend beyond the presented method and analyses, towards investigating machine unlearning as a tool for white-box interpretability. Unlearning at different granularities allows us to audit the various levels of a model's learned feature hierarchy. In this work, we illustrate how first-order, sample-wise unlearning can identify salient input features important for a single prediction. A promising future direction involves interpreting higher-level, semantically complex concepts required to learn a task or fit a data distribution, by instead unlearning a set of concept-clustered exemplars. It is also of interest to delve into how interpretability methods impose additional assumptions onto trained models, prompting questions such as how to best design and align the correct interpretability method for a given model; how to use attribution methods to compare and contrast the inductive biases of different network architectures, of models trained with robust versus non-robust objectives, of models trained using different equivariant data augmentation strategies. Further technical extensions to `UNI` include going beyond first-order approximate unlearning towards certified, second-order machine unlearning; as well as granular investigations of how the baseline definition, model's robustness and model's inductive biases exert influence on path attribution results.

## 8 ACKNOWLEDGEMENTS

This work was supported in part by the Pioneer Centre for AI, DNRF grant number P1.

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

# A APPENDIX

## A.1 VERIFYING ATTRIBUTION SPECIFICITY

To verify that `UNI` computes explanations that are specific to each task–model–input triplet, we compare its saliency attributions across models for the same image input. Visually, we observe in Figure 8 that attributions differ significantly and even reflect the inductive biases of respective models (*e.g.* grid-like artefacts are present in ViT attributions whereas smoother attributions are computed for convolutional architectures). We further present numerical results in Table 7—LPIPS (Zhang et al., 2018b) scores reflect the dissimilarity/distance between the original image and the unlearned baseline; the percentage change in confidence scores reflect how the unlearned baseline effectively reduces predictive confidence (relative to the original input).

Table 7: *UNI computes different baselines* for network architectures with different inductive biases on the same input, as seen from the drop in model confidence ($\Delta_\%$ Confidence) and image-baseline similarity scores (LPIPS$_{\text{vgg}}$, LPIP$_{\text{alex}}$).

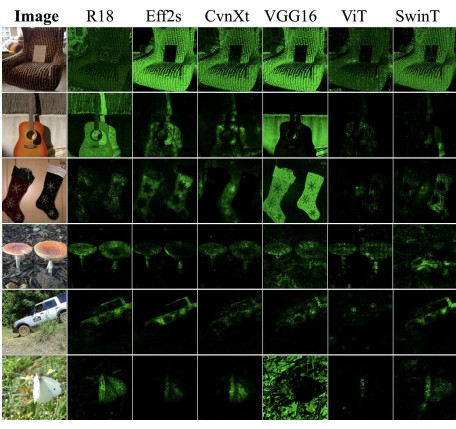

|  | $\Delta_\%$ Confidence | LPIPS$_{\text{vgg}}$ | LPIP$_{\text{alex}}$ |
|---|---|---|---|
| ResNet-18 | −82.3% | .021 ±.025 | .003 ±.005 |
| Eff-v2-s | −76.9% | .025 ±.024 | .004 ±.011 |
| ConvNeXt-T | −95.1% | .018 ±.016 | .002 ±.003 |
| VGG-16-bn | −71.6% | .017 ±.020 | .001 ±.002 |
| ViT-B-16 | −69.7% | .014 ±.015 | .004 ±.007 |
| SwinT | −84.6% | .014 ±.015 | .002 ±.002 |

Figure 8: UNI computes different attributions to explain the predictions of each model.

## A.2 PRELIMINARY RESULTS ON NLP

Table 8: *Faithfulness*: $L_2$-Distance from activation patching to attribution patching results on the residual stream (averaged over 100 samples).

|  | UNI | Random |
|---|---|---|
| Pythia-1b-v0 | **3.12** | 6.64 |
| GPT2-medium | **15.26** | 35.00 |
| Llama-3.2-1B | **5.25** | 10.17 |

We extend the testing of our method to the case of Natural Language Processing (NLP). We choose to test the application of `UNI` in the general framework of generative models (which includes classification models), and attribution of not only inputs but more generally activations. Activation patching (Heimersheim & Nanda, 2024) is one of the most widely used technique in Mechanistic Interpretability, and more generally to study the properties of LLMs' internals (Vig et al., 2020; Meng et al., 2023; Wang et al., 2022; Feng & Steinhardt, 2024; Cunningham et al., 2023; Stolfo et al., 2023; Hanna et al., 2023). This attribution method consists in analyzing a model's output variation after replacing its internal activations, following the equation:

$$\mathcal{A}_e^{\text{ACT}}(x) = F(x|e = e(x')) - F(x) \tag{4}$$

where $e$ denotes one activation in the model, $x'$ a chosen baseline, and $F$ a function of the model's output (usually the logit value of the maximum probability token of the model run on $x$).

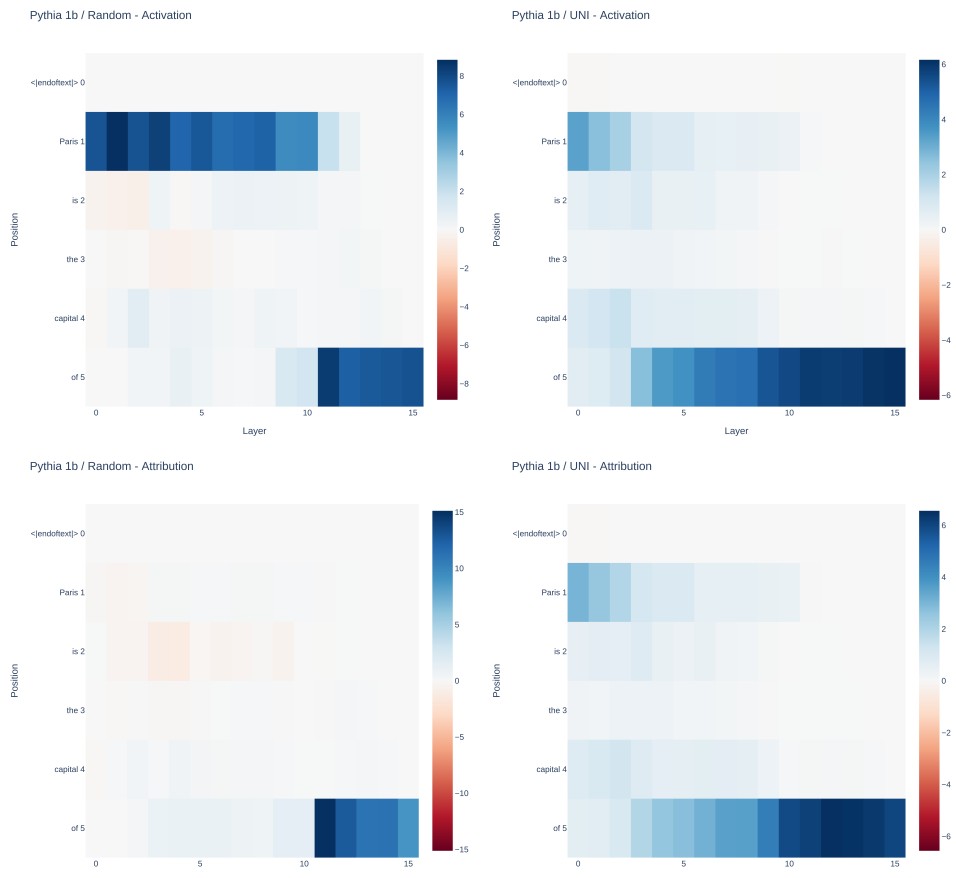

Figure 9: Visual comparison of attribution results for Activation vs. Attribution patching, with `UNI` versus Random baselines, on Pythia-1b-v0. Each cell shows the logit variation obtained by patching at a specific token and layer the residual stream of our baseline with the original activation.

Unfortunately activation patching is computationally costly, especially for purposes such as circuit discovery (Conmy et al., 2023). One of the main alternatives that solves the scalability problem is attribution patching (Nanda, 2023; Syed et al., 2023), which computes a first order Taylor approximation of Equation 4:

$$\mathcal{A}_e^{\text{ATTR}}(x) = (e(x) - e(x'))^T \nabla_e F(x') \tag{5}$$

for which the attributions for all of the activations can be computed at the same time (no patching of one single activation is performed). Despite its scalability, attribution patching suffers from a lack of faithfulness for causal interventions, mainly due to saturation and lack of linearity of the studied dependencies.

The analogy with integrated gradients seems quite striking, and indeed two recent works (to our knowledge) have tried to investigate the use of IG for more faithful attribution patching. While Marks et al. (2024) applies a very computationally complex version of such a method to small models, Hanna et al. (2024) proves the potential of IG-based attribution patching, while showing it still gets outperformed by activation patching.

We here provide a new `UNI`-based attribution method algorithm outputting faithful attributions while maintaining the scalability advantage of attribution patching. Mainly, we apply Algorithm 1 to compute a baseline $x'$ that is then used to compute Equation 1:

$$\mathcal{A}_e^{\text{UNI-ATTR}}(x) = (e(x) - e(x'))^T \nabla_e F(\texttt{UNI}(x)) \tag{6}$$

where we take $x$ to be the embedding of the input, to allow for continuous operations on it. Considering the known high faithfulness of activation patching (Hanna et al., 2024), we approximate faithuflness

of attributions computed from a baseline, as the $L_2$-distance between these attributions and the activation patching ones. The dataset used is a subset of 100 counterfactual prompts taken from Meng et al. (2023), and three different models are tested: Pythia-1b-v0 (Biderman et al., 2023), GPT2-medium (Radford et al., 2019) and Llama-3.2-1B (Dubey et al., 2024). The results can be seen in Table 8, and visuals in Figure 9. Note that no fine-tuning of UNI hyperparameters has been done, so that we expect even better results when adapting for each models. Unsurprisingly, decoding the baselines by shortest distance to the rows of the embedding matrix yields the same input, and a direct decoding of the perturbation $\delta$ doesn't provide any interesting information.

## A.3 Additional Visualisations

We supplement the main text with visualisations of the UNI baseline, attributions and path features (properties, stability and robustness). We additionally include figures elucidating the colour, texture and frequency biases post-hoc imposed by path attribution methods. From Figure 10, we observe the stability of UNI path features: our attributions can be reliably and efficiently computed with Riemann approximation. In Figures 11, 12 and 13, we present visualisations on ImageNet-C, highlighting how static choices of baselines may bias the path-attribution procedure, leading to null or noisy explanations. UNI does not impose additional post-hoc assumptions that are alien to the model's decision function. Furthermore, we present qualitative comparisons of attribution results of pre-trained models on the ImageNet-1K test set, in Figures 14, 15, 16, 17, 18 and 19. UNI attributions are visibly better localised and more semantically meaningful. Finally, we visualise the consistent, geodesic paths of monotonically increasing output confidence, discovered by UNI. As seen from Figures 20, 21, 22, 23, 24 and 25, while other path attribution methods might encounter extrema and turning points along the interpolation path from baseline to input, UNI's path features are monotonic and preserve the crucial completeness property on which the path attribution framework depends.

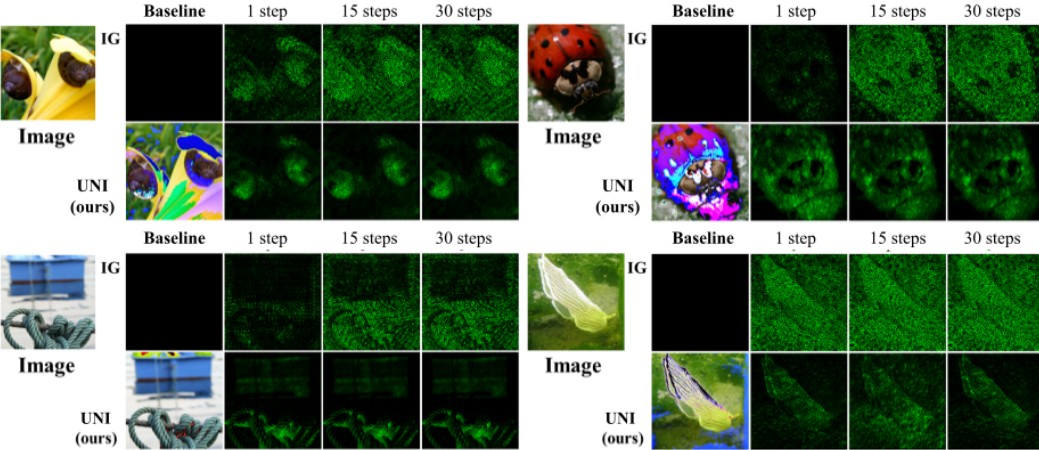

Figure 10: Comparison of attribution maps computed by Integrated Gradients and UNI, for a pre-trained ResNet-18 on the ImageNet-1K test set. UNI occludes and unlearns predictive input features; reliably localises predictive image regions; can be efficiently computed with only 1 Riemann step.

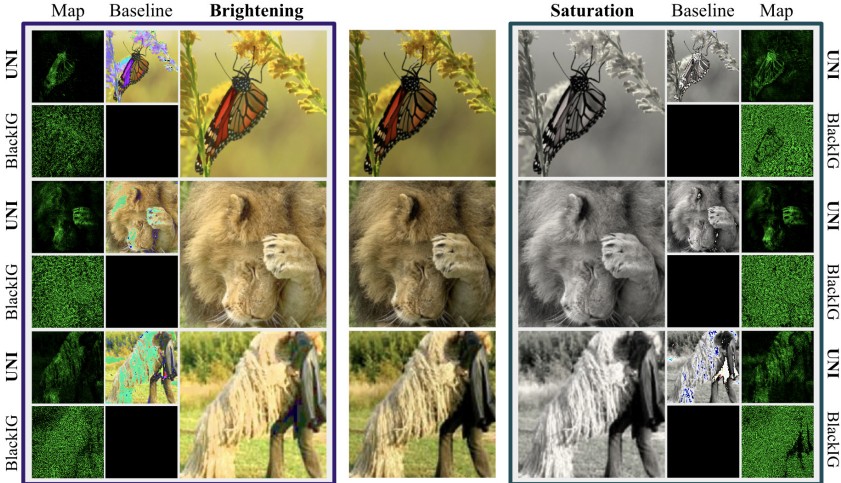

Figure 11: *Colour bias:* When an image's brightness or saturation is altered, IG with a black baseline fails to identify dark features, such as the wings of the butterfly (R1) or black jacket (R3).

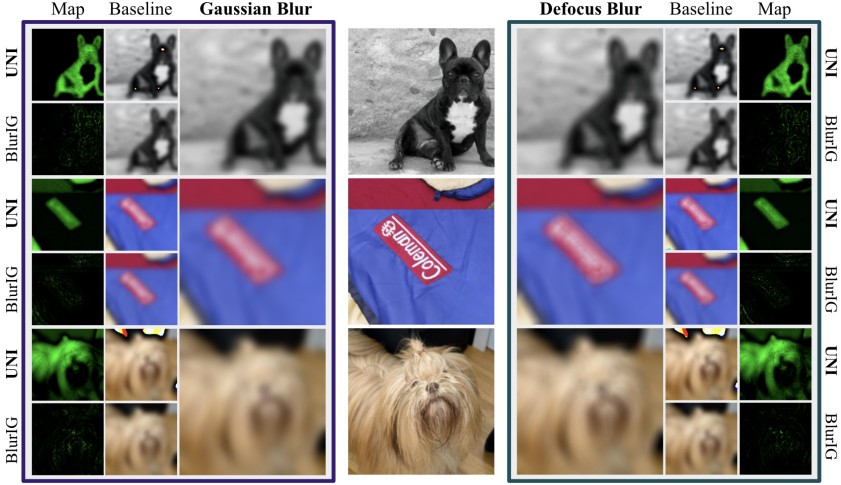

Figure 12: *Texture bias:* Using a blurred baseline for IG leads to a smoothness assumption in image texture, which leads to missingness in attribution when the input is also gaussian or defocus blurred.

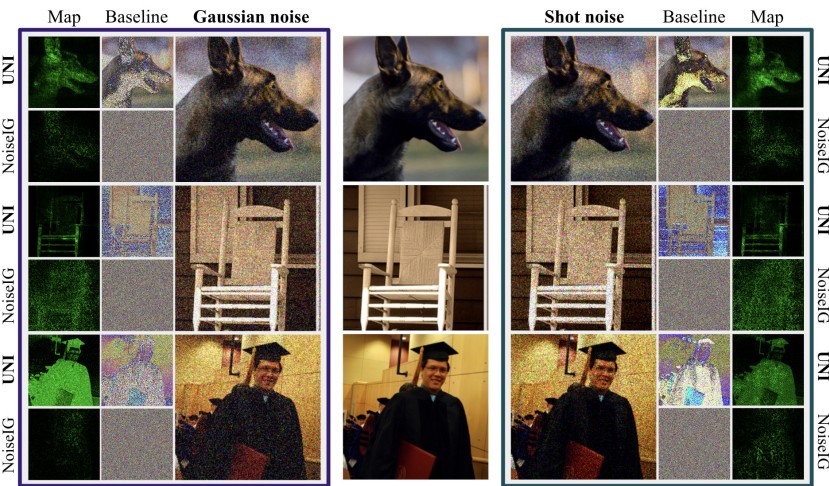

Figure 13: *Frequency bias:* A gaussian noised baseline for IG renders it vulnerable to high-frequency corruptions. Adding gaussian or shot noise to the image yields unmeaningful, noisy attributions.

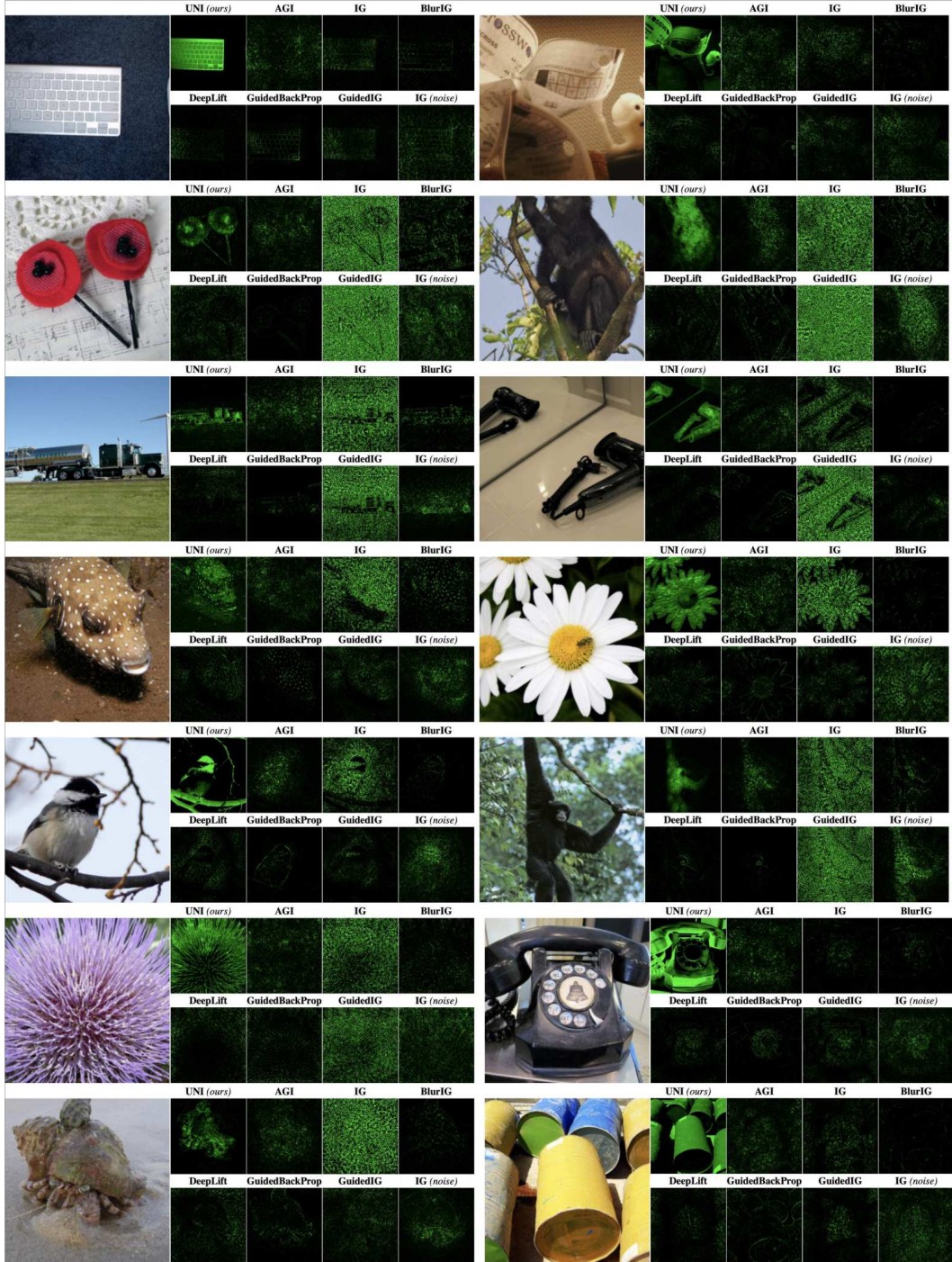

Figure 14: *Comparing attributions (ResNet-18):* `UNI` attributions demonstrate higher saliency, fidelity and faithfulness relative to conventional baselines on the ImageNet test set.

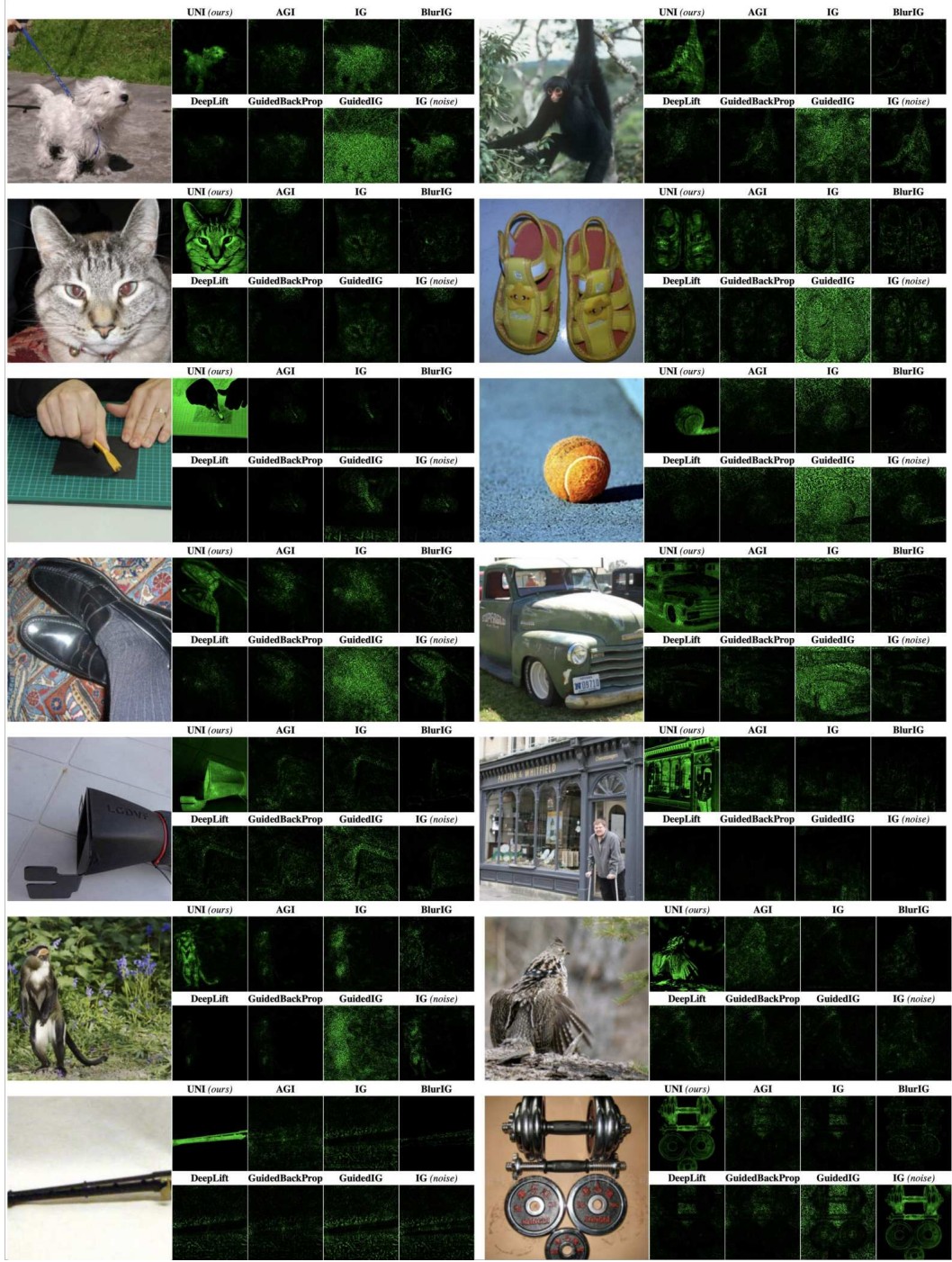

Figure 15: *Comparing attributions (EfficientNet-v2-small):* UNI attributions demonstrate higher saliency, fidelity and faithfulness relative to conventional baselines on the ImageNet test set.

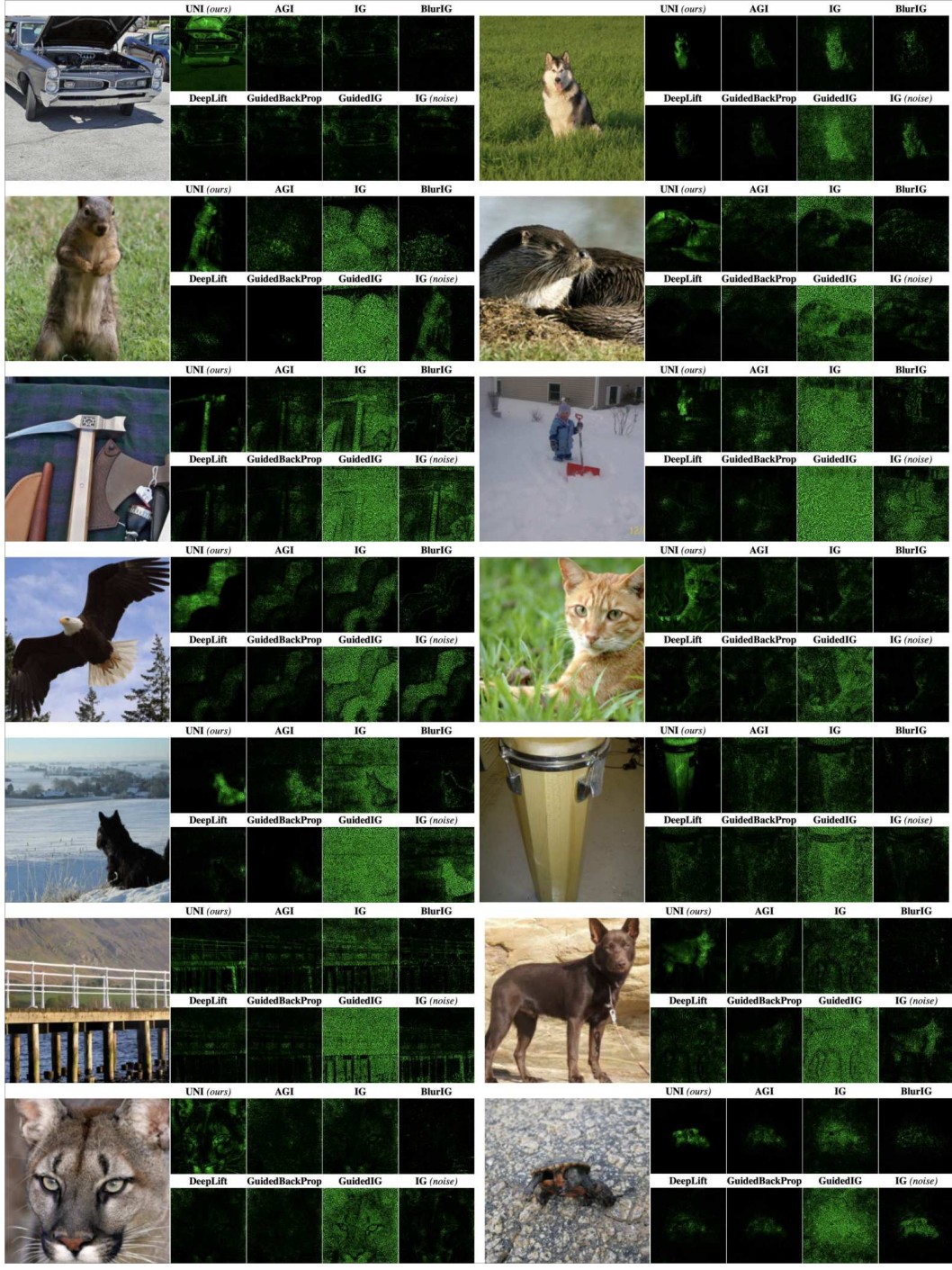

Figure 16: *Comparing attributions (ConvNeXt-Tiny):* UNI attributions demonstrate higher saliency, fidelity and faithfulness relative to conventional baselines on the ImageNet test set.

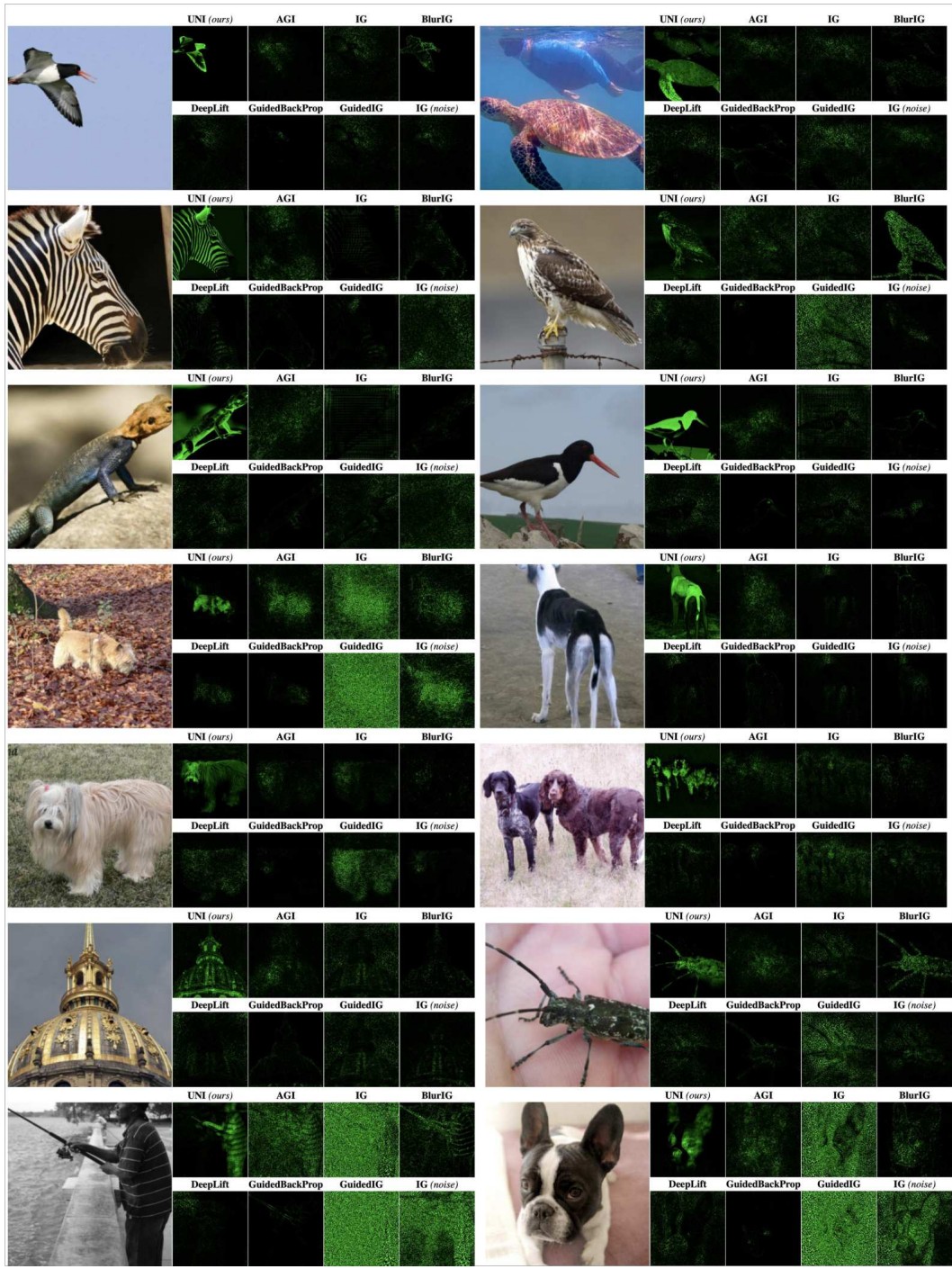

Figure 17: *Comparing attributions (VGG-16-bn):* `UNI` attributions demonstrate higher saliency, fidelity and faithfulness relative to conventional baselines on the ImageNet test set.

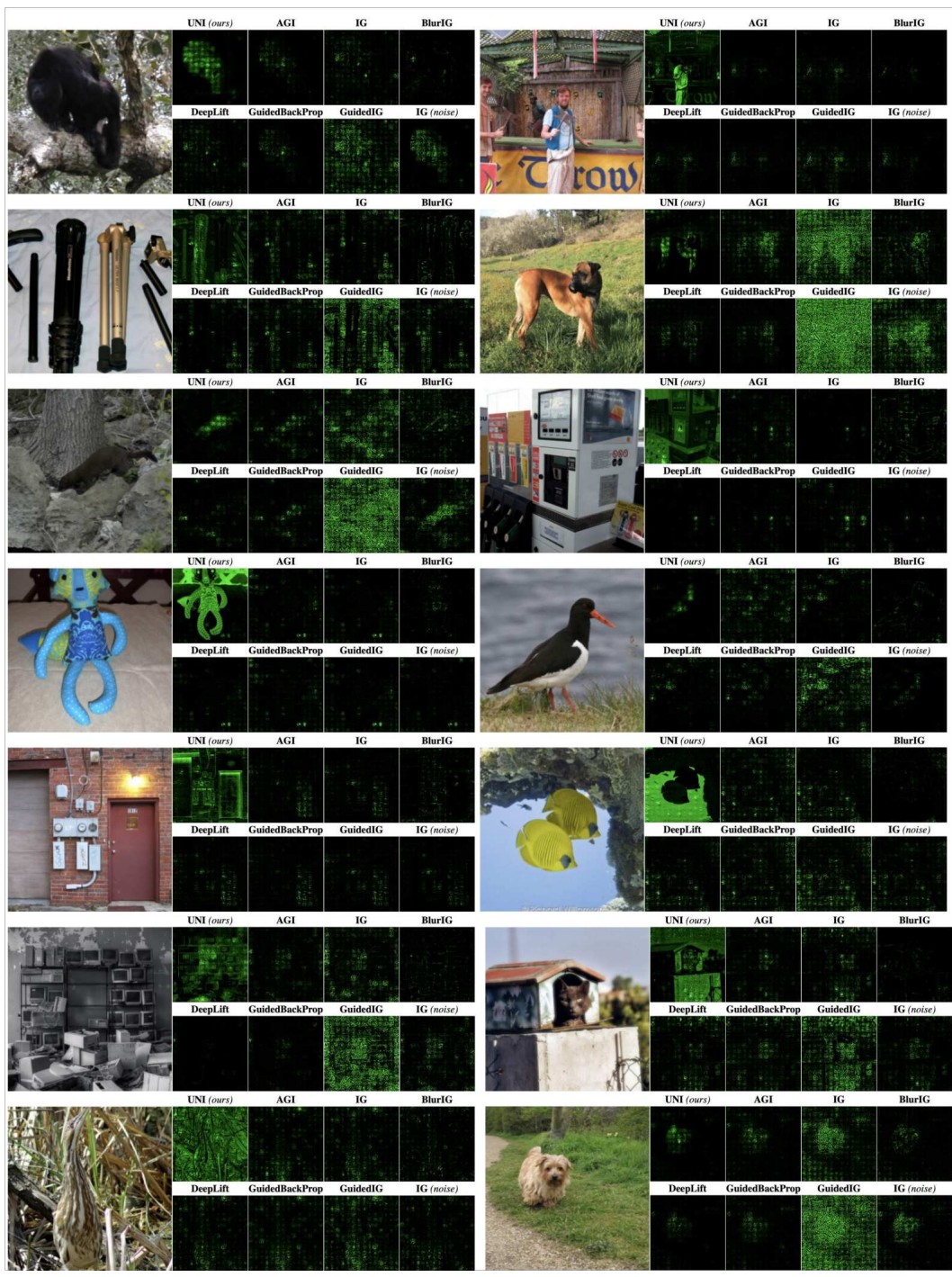

Figure 18: *Comparing attributions (ViT-B_16):* UNI attributions demonstrate higher saliency, fidelity and faithfulness relative to conventional baselines on the ImageNet test set.

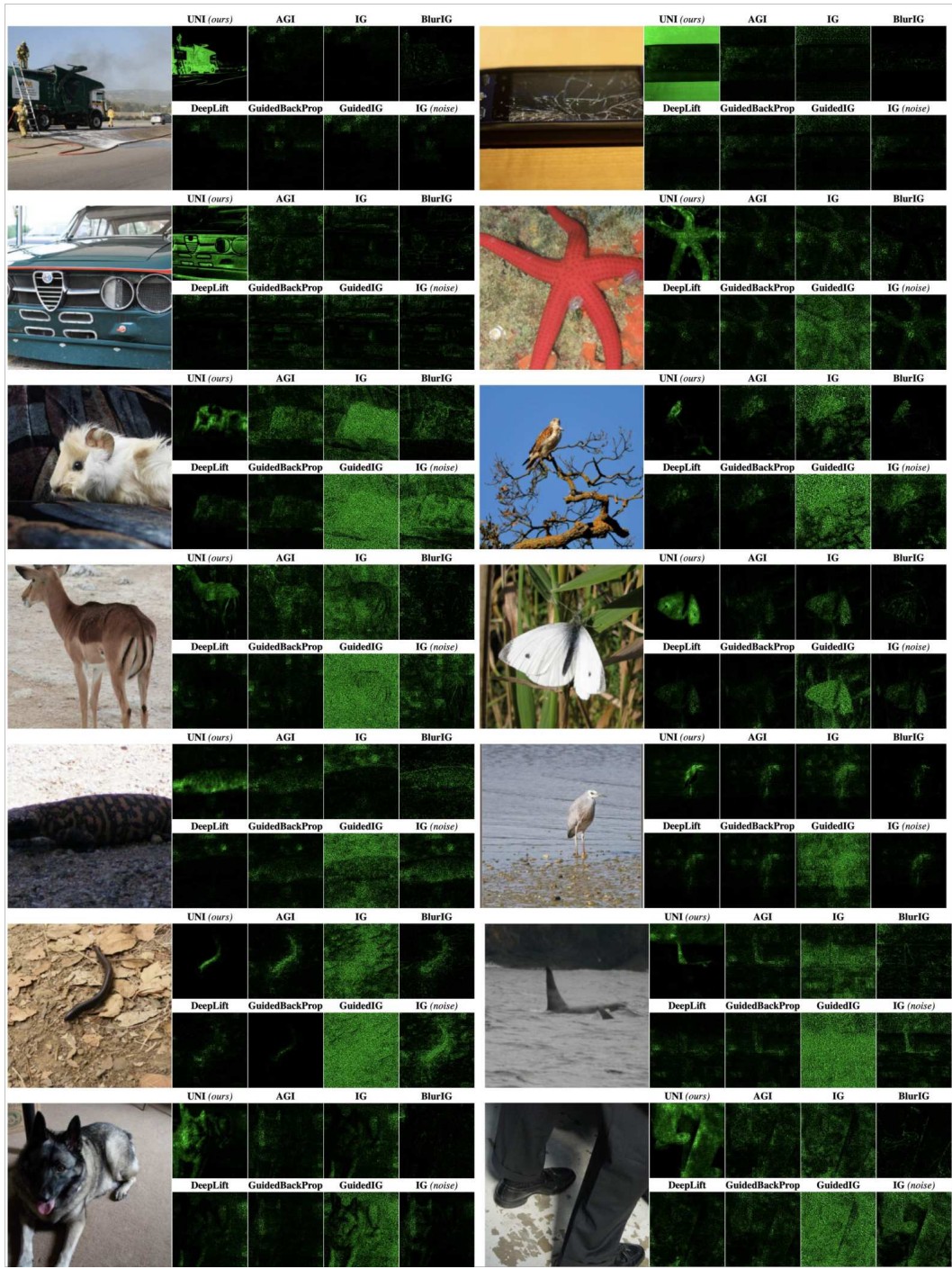

Figure 19: *Comparing attributions (Swin-Transformer-Tiny):* `UNI` attributions demonstrate higher saliency, fidelity and faithfulness relative to conventional baselines on the ImageNet test set.

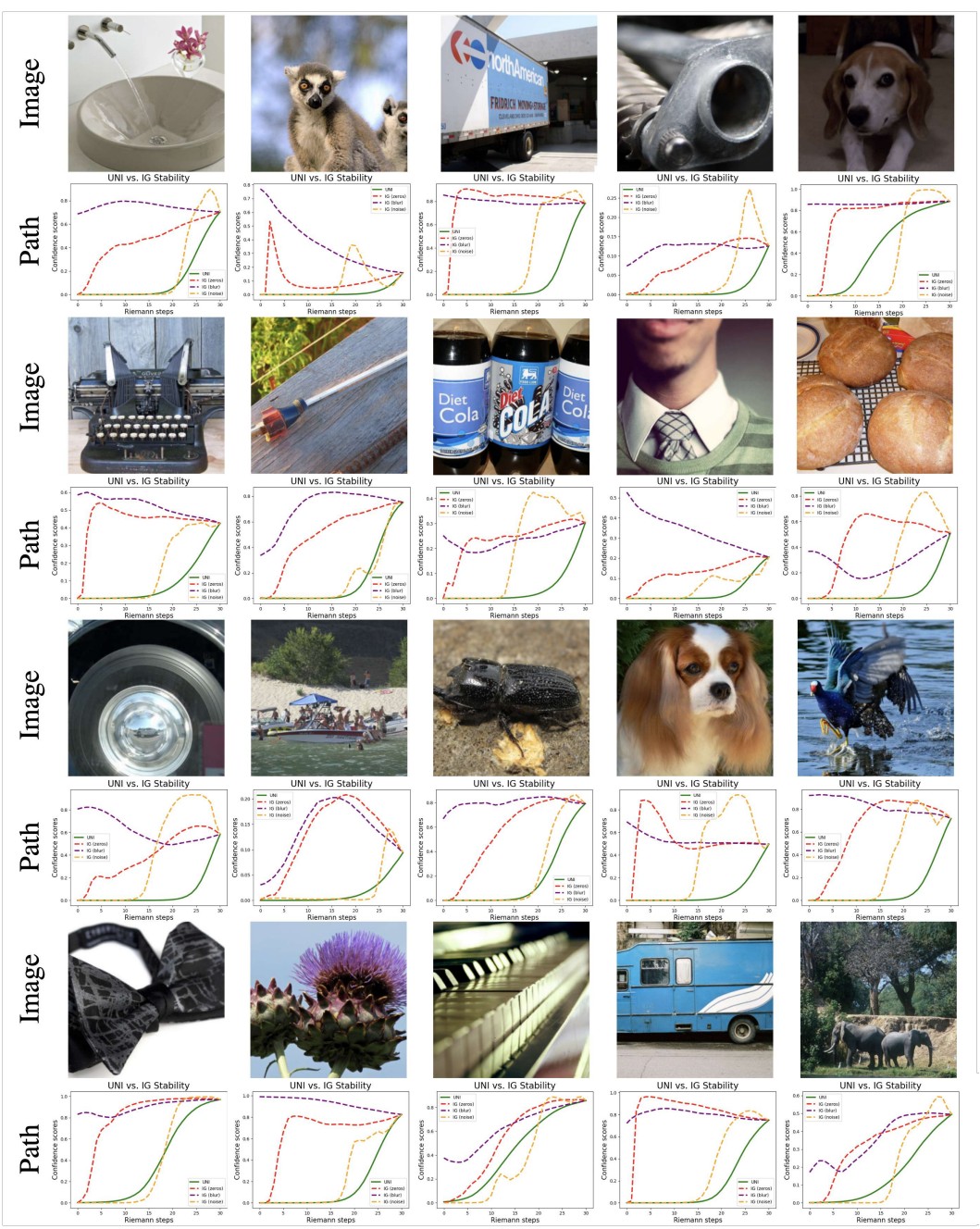

Figure 20: *Comparing paths (ResNet-18):* UNI discovers geodesic paths of monotonically increasing output confidence, preserving the completeness property required for robust attributions.

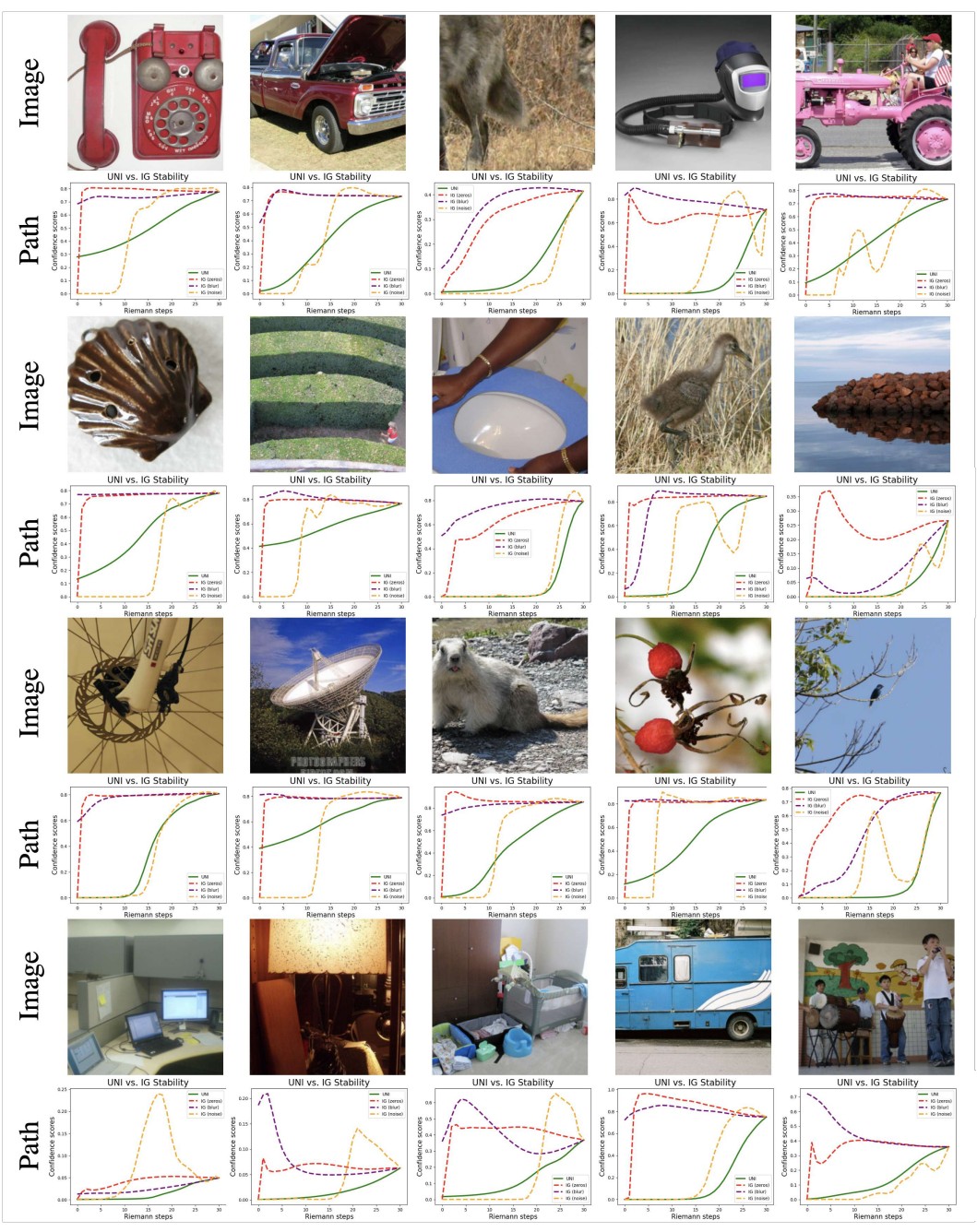

Figure 21: *Comparing paths (EfficientNet-v2-small):* `UNI` discovers geodesic paths of monotonically increasing output confidence, preserving the completeness property required for robust attributions.

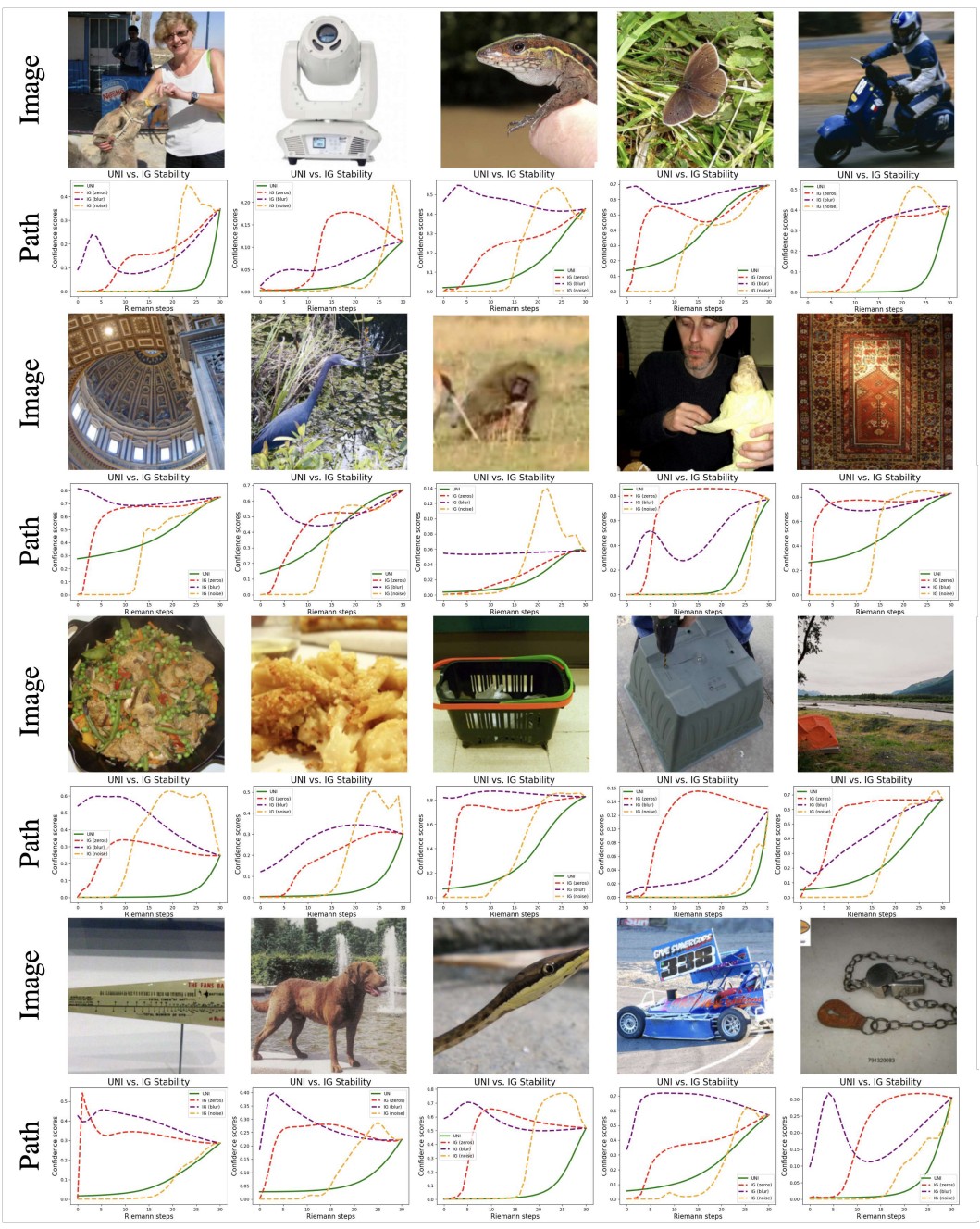

Figure 22: *Comparing paths (ConvNeXt-Tiny):* `UNI` discovers geodesic paths of monotonically increasing output confidence, preserving the completeness property required for robust attributions.

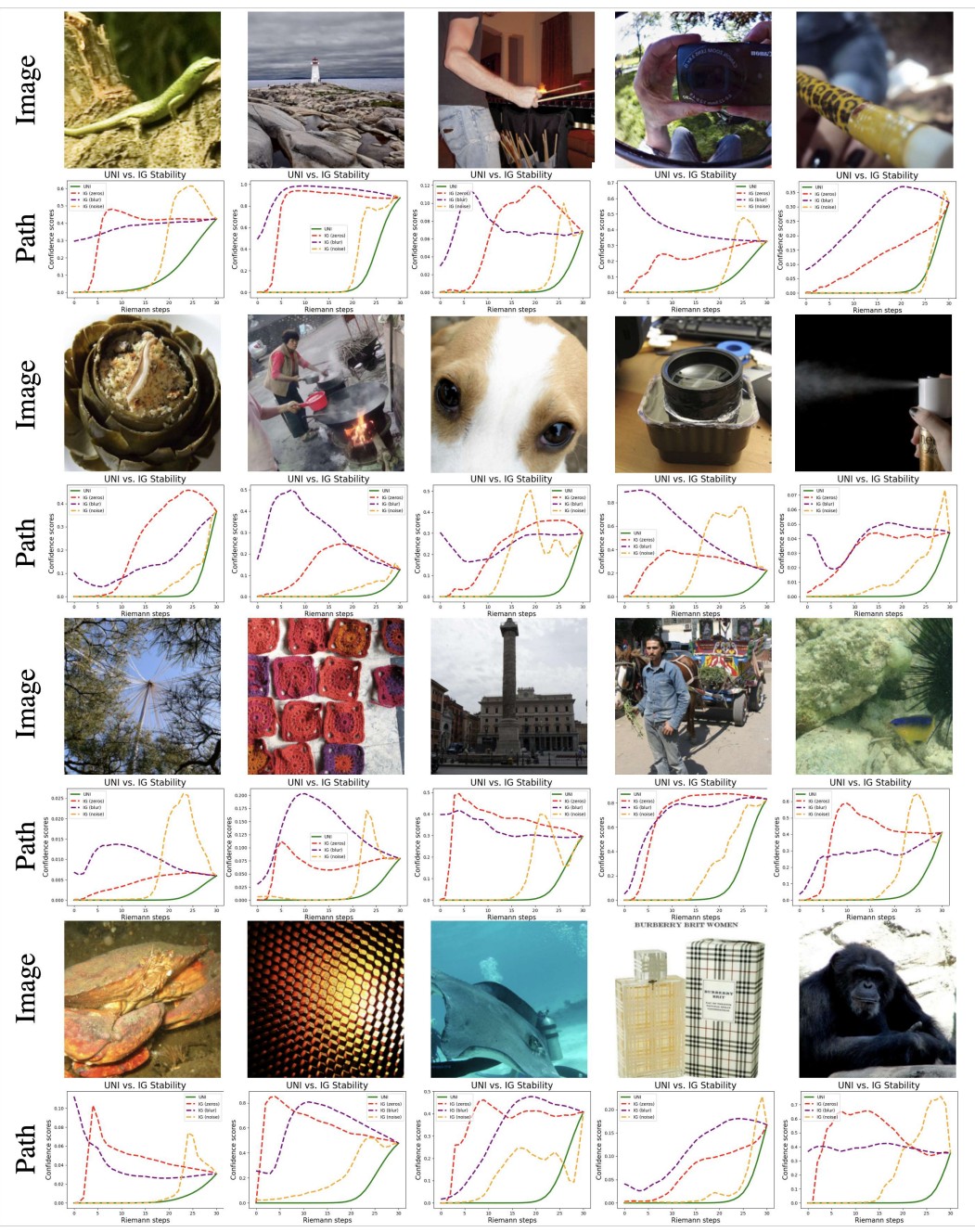

Figure 23: *Comparing paths (VGG-16-bn):* UNI discovers geodesic paths of monotonically increasing output confidence, preserving the completeness property required for robust attributions.

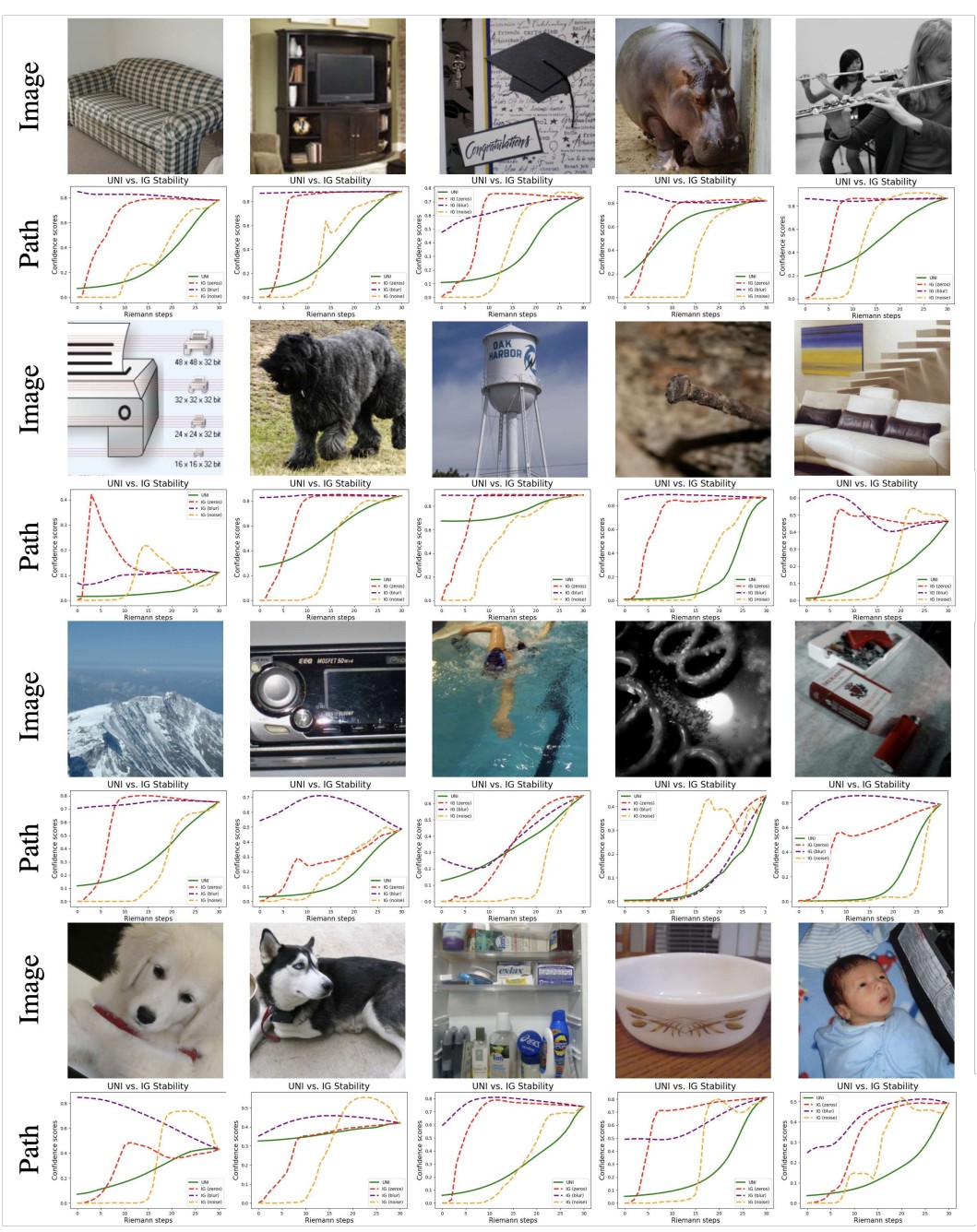

Figure 24: *Comparing paths (ViT-B_16):* UNI discovers geodesic paths of monotonically increasing output confidence, preserving the completeness property required for robust attributions.

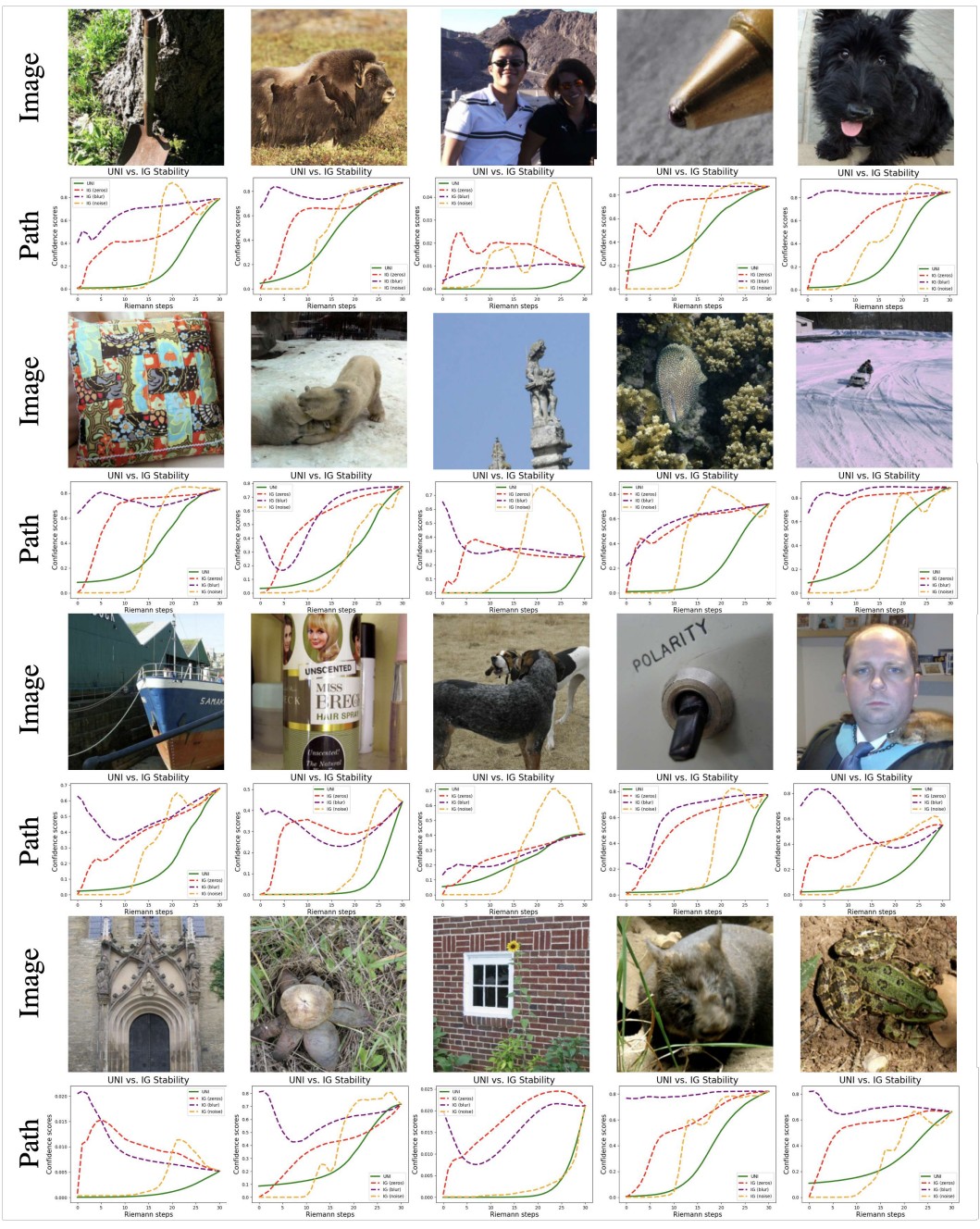

Figure 25: *Comparing paths (Swin-Transformer-Tiny):* UNI discovers geodesic paths of monotonically increasing output confidence, preserving the completeness property required for robust attributions.

