# OpenReview forum: "Unlearning-based Neural Interpretations"
_ICLR.cc/2025/Conference — ICLR 2025 Oral_

### Official Review · Reviewer_rhWZ · 2024-10-30

**Soundness:** 3
**Presentation:** 3
**Contribution:** 3
**Rating:** 6
**Confidence:** 4

**Summary:**

This paper studies the specific problem of baseline selection in feature attribution. The authors argue that the optimal baseline choice depends on the task-model-input combination and propose an unlearning-based approach to improve baseline selection. By combining the baseline selection method with a well-known path method — IG, the resulting attribution method UNI demonstrates improved performance in various aspects according to the experimental results.

**Strengths:**

1. The paper is clearly motivated and easy to follow.
2. The authors collect a set of desired properties for the integration path, which could benefit following work on baseline selection.
3. The experiments are carefully designed, thoroughly evaluating the proposed method from different perspectives that support the authors’ claims.

**Weaknesses:**

1. The description of the proposed method is somewhat oversimplified, raising concerns about the overall depth of the work. It would benefit from further elaboration and discussions on each step in Algorithm 1.
2. The reasoning behind why activation-matching ensures the removal of all salient features is unclear.
3. Further clarification on some technical details about monotonicity would be beneficial. Please refer to questions 3, 4, and 5 for specific points.
4. There are some minor errors and formatting issues, for example:
    - Different formats of $\varepsilon$ in Algorithm 1.
    - Citation wrapped by parenthesis as part of a sentence on line 261; repeated parentheses on line 269.
    - Line 349 refers to "Figure 3", which appears to be a mislabeling of "Table 3".

**Questions:**

1. Could the authors elaborate on why activation matching ensures the removal of salient features? Furthermore, given an activation vector $F(x;\hat{\theta})$, there may be multiple matches in the decision space within the budget $\varepsilon$. Is there any guarantee of the convergence for this matching, or is it based solely on empirical observations?
2. The goal of the unlearning is to derive $F(x;\hat{\theta})$ for matching. What advantage does it offer over directly adopting a non-informative activation, say $(\frac{1}{d_Y},\ldots,\frac{1}{d_Y})$? Could the latter be even more powerful as it removes features that are informative in general?
3. Could the authors clarify the meaning of $F_c(x_i)$ at lines 297-298 mean? It is unclear how the model processes a single feature $x_i$ instead of an instance $x$.
4. How are the two conditions in lines 297-298 satisfied?
    - Assuming the authors intended $F_c(x)$ rather than $F_c(x_i)$, the monotonicity of the class confidence does not necessarily imply a consistent sign of the derivatives w.r.t. each feature $x_i$. This indicates that the sign of $\nabla_{\tilde{x}_i}F_c(\tilde{x})$ may vary along the path, which contradicts the claim of the first condition.
    - Solving the gradient saturation problem is one motivation for IG. However, if a feature $x_i$ has a saturated gradient at $x$, i.e. $\nabla_{x_i}F_c(x)\approx0$, the second condition becomes problematic as it requires the derivate of $x_i$ to remain 0 over the integration path, leaving gradient saturation unsolved.
5. Could the authors clarify the relationship between completeness and path monotonicity? Completeness, as given by the gradient theorem, should be path-independent.
6. Many attributions provided by UNI closely align with the details of the input images. To demonstrate that the explanations are not simply grayscale reproductions of the inputs, could the authors provide example attributions of different models on the same inputs to show whether they distinguish from each other (under the assumption that different models behave differently)?

---

> ### Author Response · Authors · 2024-11-24
> **Author Response**
>
> We thank the reviewer for their detailed feedback and constructive suggestions.
>
> **Re: W1 -** We strengthen our discussion of the motivation and technical details behind UNI with a simple motivating example using a gaussian mixture model (Section 4.3 and general response). We explain the necessity of unlearning and activation matching, by illustrating how a naive gradient descent approach (to minimise model confidence) omits more desirable baseline solutions that UNI is able to discover. We observe that UNI preserves path monotonicity, low-curvature and proximity and justify how it satisfies the weak dependence property for conformal path attribution. We further note that UNI has in-depth contributions to broader XAI, and elaborate on this in the general response and revised manuscript. In particular, we are optimistic that UNI’s success will facilitate further research into 1) granular unlearning for hierarchical feature/concept attribution; 2) aligning XAI methods and models; 3) comparing network inductive biases and interpretability during training.
>
> **Re: W2 & Q1 -** To motivate activation matching, first consider that ReLU networks have decision boundaries that are well represented as piecewise linear functions. First-order unlearning reduces the magnitude of piecewise linear weights that are highly activated by salient features; activation matching discovers a baseline in the image space that replicates this activation pattern. This encourages the baseline and the input to lie in the same connected component, which crucially satisfies the weak dependence property for conformal path-based attribution.
>
> **Re: W3 & Q3-5 -** We revise Sections 4.2-4.3 to clarify concepts of weak dependence and completeness and how they relate to the quality of the path features (monotonicity, low curvature, proximity). Weak dependence—a smooth transition between the absence and presence of salient features—is crucial for meaningful path attribution, and is only compatible with completeness in the case where the baseline and the input lie in the same connected component (in the case of piecewise-linear models). A path (from baseline to image) with monotonically increasing model confidence suffices to satisfy a weak version of the criteria for valid path features $\textrm{sgn}(\nabla_x F_c(x)) \cdot \textrm{sgn}(\nabla_{\tilde{x}} F_c(x^\prime))=1$ (Akhtar & Jalwana, 2023); a path that exhibits low curvature (i.e. eigenvalues of the hessian) on and in a neighbourhood of the baseline reduces the variability and robustness of the calculated gradients under small-norm perturbations; a specific baseline (enforced by step 5 of Algorithm 1) that lies within an $\varepsilon$-ball of the image satisfies the proximity condition required for weak dependence.
>
> **Re: W4 -** Fixed and thanks!
>
> **Re: Q2 -** An unlearned baseline is necessary because it reflects which features are salient for a specific model’s prediction. Static baseline choices (e.g. using constant (black, channel mean) pixel baselines, gaussian blur, random baselines) project assumptions onto the model, which result in undesirable biases that confound interpretation results. Since it is important to attribute features in the pixel space, the suggested baseline could 1) introduce a colour bias that specific pixel values are unimportant for inference, 2) lie far away from the local image neighbourhood and violate the weak dependence property. Unlearning allows for conformal paths from baseline to input, facilitating robust interpretations.
>
> **Re: Q6 -** To demonstrate that UNI attributions align well with specific task-model-image triads, we perform additional experiments in Appendix A.1. We visually and numerically compare UNI attributions across models for the same image input, with the LPIPS image similarity metric. We observe in Figure 8 that attributions differ significantly and even reflect the inductive biases of respective models (e.g. grid-like artefacts are present in ViT attributions whereas smoother attributions are computed for convolutional architectures). Table 7 scores reflect the dissimilarity between the original image and unlearned baselines (pixel-wise and in terms of model confidence).
>
> We again express our gratitude to the reviewer for their help in improving our work. We look forward to a fruitful discussion!

---

> ### Comment · Reviewer_rhWZ · 2024-11-25
>
> Thank you for the detailed response, which addresses most of my concerns. I have raised my score from 5 to 6.

---

### Official Review · Reviewer_Wrfy · 2024-11-03

**Soundness:** 3
**Presentation:** 3
**Contribution:** 2
**Rating:** 8
**Confidence:** 4

**Summary:**

The authors introduces a novel approach to enhance gradient-based post-hoc attributions.
The authors argue that static baselines (e.g., black images, blurred noise) impose unintended biases on attribution maps, leading to fragility and unfaithful interpretations. The proposed UNI framework generates a dynamic baseline by perturbing the input in an unlearning direction of steepest ascent, theoretically optimizing for a featureless reference that aligns with the model's predictive function.
This adaptive baseline improves the path attribution by reducing curvature and enhancing robustness, demonstrated empirically through comparisons on benchmark datasets like IN and various baseline models.

**Strengths:**

1. **Innovative Baseline Approach**: The idea of using unlearning to create adaptive baselines is novel and addresses a known limitation in static baseline approaches.
2. **Robustness and Faithfulness**: The empirical results, including MuFidelity scores and robustness to adversarial perturbations, highlight the potential of the method to produce more reliable attributions.
3. **Comprehensive Experiments**: The paper includes evaluations on multiple models and datasets, adding credibility to the proposed approach.

**Weaknesses:**

However, I spotted some problems, some major (**M**) and minor (**m**):

**M1.** Literature Gaps: the related work section omits significant prior research on black-box interpretability (RISE, Sobol...) and miss some really important faithfulness metrics. Also a paper from last year that could be relevant (Saliency strike back) could be discussed.

**M2.** Faithfulness Metrics: while the paper discusses MuFidelity as a metric, it does not consider complementary metrics such as deletion and insertion, which could offer a broader validation of faithfulness and robustness. Including these would provide a more comprehensive evaluation of the method's performance. Also MuFidelity contain many Hyperparameter that are known to favorise certain method, could you state how did you approximate MuFidelity (number of sample)  and also the value of $|S|$ ?

**M3.** Theoretical Justification: the paper lacks a theoretical discussion on how UNI would perform under linear cases or simple functions. Providing proofs or theoretical insights for these cases would enhance the paper's depth and foundational strength.

**M4.** Broader Impact on Understanding Neural Networks: The practical implications of this method for gaining new insights into neural networks remain unclear. How much new knowledge / what have learn with this method that we haven't with other attribution methods ?

Now for the minor problem:

**m1.** Limited Baseline Comparisons: the paper mainly contrasts UNI with IG and a few other attribution methods. Expanding the baseline comparison to include more recent methods (including black-box one such as RISE or HSIC that are know to have good results on faitfhulness metrics) and varied methods would strengthen the experimental section.

**m2.** Minor Typos: I havent found a lot of typos, just line 468: "simultatenously".

**Questions:**

- Could you adapt the method to a low dimensional subspace path (eg delta is parametrized by 7x7 mask that we interpolate and then optimize?)
- How would the proposed method adapt if applied to models beyond image classification, such as language models or multimodal networks?
- Could the authors consider a variant of UNI that leverages multi-scale features for tasks involving more intricate hierarchical dependencies?


**Overall, I think the paper is well-executed with thorough analysis and comprehensive results. While the contribution may be of limited impact within the broader field of explainable AI (see Major issues), I tend to give a marginal accept.**

---

> ### Author Response · Authors · 2024-11-23
> **Author Response**
>
> We thank the reviewer for their detailed feedback and insightful comments.
>
> **1. Major comments**
> **Re: M1 -** We thank the reviewer for their valuable suggestions. RISE, Sobol and FORGrad (Saliency strikes back) are indeed very relevant to our work and we discuss their contributions in Section 6, particularly how they improve attribution faithfulness.
>
> **Re: M2 -** We supplement MuFidelity score results with deletion and insertion score results in Tables 3 and 4, and discuss how they provide an approximately causal view into which pixels are necessary and sufficient for a particular prediction result. We also detail hyperparameters used (for MuFidelity, deletion and insertion scores) in Section 5.1.
>
> **Re: M3 -** We strengthen the theoretical arguments of UNI with a simple motivating example using a gaussian mixture model, presented in Section 4.3 of the revision (and the general response). We explain the necessity of unlearning and activation matching, by illustrating how a naive gradient descent approach (to minimise model confidence) omits more desirable baseline solutions that UNI is able to discover. We observe that UNI preserves path monotonicity, low-curvature and proximity and justify how it satisfies the weak dependence property for conformal path attribution.
>
> **Re: M4 -** We thank the reviewer for their thought-provoking question. We discuss the broader contributions of UNI to XAI both in the general response and the revised paper manuscript. In particular, we are optimistic that UNI’s success will facilitate further research into 1) granular unlearning for hierarchical feature/concept attribution; 2) aligning XAI methods and models; 3) comparing network inductive biases and interpretability during training.
>
> **2. Minor comments**
> **Re: m1 -** While this work mainly focuses on white-box attribution methods, since black-box methods typically rely on different mechanisms (repeated forward passes instead of gradient information). We welcome the suggestion and have included a discussion of black-box methods in Section 6.
>
> **Re: m2 -** Fixed and thanks.
>
> **3. Questions**
> **Re: Q1 -** We have not experimentally considered low-dimensional subspace paths, though this could potentially further improve the efficiency of UNI path attribution.
>
> **Re: Q2 -** Under the reviewer’s suggestion, we expand UNI experiments to the NLP domain and report promising results on activation patching for interpreting language models (Appendix A.2).
>
> **Re: Q3 -** This is an intriguing question which ties in with the UNI unlearning mechanism. We elaborate on this in both the revised manuscript and the general response:
> > UNI shows that first-order, sample-wise unlearning identifies and removes salient features predictive for a single sample. An exciting next step involves unlearning at the distribution level to localise and erase higher level concepts, such as unlearning a set of feature-clustered exemplars to erase common concepts. As noted by Reviewer Wrfy, unlearning in conjunction with activation matching can attribute features hierarchically, from local image textures, to intermediate shape prototypes, to class-semantic concepts.
>
> We again express our gratitude to the reviewer for helping improve our work and look forward to a fruitful discussion!

---

> ### Comment · Reviewer_Wrfy · 2024-11-24
>
> Dear Authors,
>
> Thank you for your detailed response.
>
> I really appreciated the additional metrics which I think strengthens the paper. The Gaussian Mixture Model example adds much-needed theoretical clarity, excellent idea.
> Your discussion of UNI’s broader impact highlights its potential to inspire future work.
>
> I’ve increased my score from 6 to 8 after reviewing your revisions. Good luck with the acceptance!

---

### Official Review · Reviewer_G3s3 · 2024-11-04

**Soundness:** 4
**Presentation:** 4
**Contribution:** 4
**Rating:** 10
**Confidence:** 4

**Summary:**

The authors identify that current baseline approaches in path attribution methods (like Integrated Gradients) inject harmful assumptions about color, texture, and frequency that can lead to biased and fragile attributions. UNI proposes computing baselines by perturbing inputs in an "unlearning" direction, to erase salient features while maintaining model-specific properties. The method demonstrates improved faithfulness, robustness, and stability compared to existing approaches across multiple model architectures and datasets.

**Strengths:**

The paper offers a well-structured analysis of an important problem in neural network interpretability. The authors identify a specific, previously overlooked issue: how static baseline approaches in attribution methods can introduce unintended biases in three distinct categories - color (shown through experiments with brightness/saturation changes), texture (demonstrated via gaussian/defocus blur tests), and frequency (validated through gaussian/shot noise experiments).

This observation is methodically supported through experiments across a comprehensive set of modern architectures, including both CNN-based models and transformer-based models. Their UNI method shows measurable improvements in three key areas: faithfulness (with MuFidelity scores showing consistent improvements across architectures), robustness (demonstrated through FGSM adversarial attacks, achieving higher Spearman correlation coefficients), and stability (shown through detailed Riemann approximation experiments that work even with just one step).

The theoretical foundation includes careful mathematical analysis where they provide explicit bounds on Riemann approximation error and prove how their approach maintains monotonicity and completeness properties. The extensive experimental results in the appendix, spanning all (20+!) detailed figures, provide clear visual evidence of how their unlearning-based approach improves upon traditional static baselines across different types of images and corruptions.

Well done!

**Weaknesses:**

Largely this is a well written paper, here are a few potential avenues for improvement :
- Would it be possible to make stronger theoretical guarantees about the optimality of the unlearned baseline?
- Image classification baselines are pretty standard, I wonder how these baselines will look like in other domains such as NLP and audio.
- There could be some more discussion around hyperparameters, an often overlooked aspect in explainability literature
- It is mentioned that other techniques are inefficient, a detailed analysis of computational overhead might solidify contributions more strongly in that aspect.

**Questions:**

No particular questions

---

> ### Author Response · Authors · 2024-11-24
> **Author Response**
>
> We thank the reviewer for their insightful feedback and thorough understanding of our work.
>
> **Re: W1 -** We strengthen the theoretical arguments of UNI with a simple motivating example using a gaussian mixture model, presented in Section 4.3 of the revision (and the general response). We explain the necessity of unlearning and activation matching, by illustrating how a naive gradient descent approach (to minimise model confidence) omits more desirable and optimal baseline solutions that UNI is able to discover. We observe that UNI preserves path monotonicity, low-curvature and proximity and justify how it satisfies the weak dependence property for conformal path attribution.
>
> **Re: W2 -** Under the reviewer’s recommendation, we expand UNI experiments to the NLP domain and report promising results on activation patching for interpreting language models (Appendix A.2). Equation 6 shows how UNI’s underlying principle of an unlearned point-of-reference can be adapted for an activation patching baseline. We report interpretability results on counterfactual prompts – we observe that like in the image domain, UNI improves the faithfulness of attributions for the 3 examined generative text models (Pythia-1b-v0, GPT2-medium, Llama-3.2-1B).
>
> **Re: W3 -** To supplement the hyperparameter details given in Section 5, we discuss how crucial hyperparameters for explainability metrics (MuFidelity, deletion and insertion scores) are chosen for fair comparison in Section 5.1. Indeed, certain hyperparameters (e.g. |S| in MuFidelity score) have considerable impact on the signal-to-noise ratio of attributions, we provide justifications of hyperparameter settings and references to existing norms.
>
> **Re: W4 -** This is an astute point. We incorporate discussions on how considerations of computational cost have influenced design choices of the UNI algorithm. Notably, UNI relies on a first-order method for unlearning, even though higher-order methods which leverage Hessian information should give better optimality bounds. However, computing the Hessian is expensive and error contributions of higher order terms quickly become negligible, which yields diminishing returns for high-order unlearning methods. Other design choices which impact performance include using only 1 unlearning step (with a large enough unlearning step size $\eta$), using fewer PGD activation matching steps $T$ for a sufficiently large PGD step size. Finally, since UNI computes stable, monotonic and conformal path features, it requires fewer Riemann approximation steps for path attribution, which drastically improves efficiency.
>
> We again express our gratitude to the reviewer for their encouragement and help in improving our work. We look forward to a fruitful discussion!

---

### Official Review · Reviewer_nTJj · 2024-11-04

**Soundness:** 3
**Presentation:** 4
**Contribution:** 3
**Rating:** 8
**Confidence:** 4

**Summary:**

This paper suggests a method for creating references for IG and other backpropagation-based explanation models. The reference is based on an untrained model with perturbations applied to the original image with respect to the model. The intuition, insights, and experimental results are presented as evidence.

**Strengths:**

Authors well-summarize the previous search, and clearly explain the required properties needed for the reference. The intution is clearly written, and well-evidenced by experimental results.

**Weaknesses:**

1. Even though they effectively present their intuition, there is no theoretical justification for their method. In particular, the pseudo-code is not explained enough. Additionally, it is unclear why the authors chose to implement it in the way they described in the code. The method itself needs further elaboration.

2. The suggested method is limited to explainable methods that require a reference, which restricts the broader applicability of the paper.

**Questions:**

1. The images shown in the figure may suggest that the white color appears relatively weaker compared to the black color. Is there any bias towards the white color? If not, why is the gray part of the swimsuit not highlighted? In Figure 2, the white swan also shows a similar tendency. Do you have any qualitative measures for this?

2. Why do you use KL divergence? Have you tried different methods as well?

---

> ### Author Response · Authors · 2024-11-23
> **Author Response**
>
> We thank the reviewer for their feedback and thoughtful comments.
>
> **Re: W1 -** We provide an illustrative example of UNI’s behaviour on a gaussian mixture model in Section 4.3 to clarify design choices of the UNI algorithm.
>
> **Re: W2 -** We demonstrate that UNI is applicable beyond path-based image attribution methods by providing supplementary results in NLP. In particular, we adapt UNI to interpret generative language models, improving the stability and fidelity of recent activation patching methods in Appendix A.2. We observe that UNI’s underlying principle of finding an unlearned point-of-comparison is well aligned with XAI tasks: the unlearning mechanism lends itself to performing interventions on salient features, while activation patching constructs data probes for counterfactual model behaviour. Motivated by reviewers’ comments, we further discuss exciting follow-up questions prompted by UNI and its broader contributions to XAI in Section 7.
>
> **Re: Q1 -** While the saliency scores attributed to image features is dependent on the model’s decision function (and the input sample), we believe that in Figure 2’s example, the black swan could be identified as being more predictive of the class label because of the animal’s prototypical, swan-like shape and because it is closest to the foreground. For the case of the swimsuit (now Appendix Figure 7), a likely reason is that models have been known to disproportionately rely on object boundaries for classification. While it appears that UNI does attribute greater salience to the lower body of the swimsuit, it is able to also trace the outlines of the upper body and provide a higher fidelity attribution than other baselines.
>
> **Re: Q2 -** We leverage KL divergence to match the learned model’s output probabilities to the unlearned model’s probabilities. The KL divergence is particularly applicable because it is directed and asymmetric, which is important for activation matching in baseline construction. We have considered other objectives, such as simply using gradient descent to minimise model confidence and taking the minimiser as the baseline. However, as illustrated in Figures 6a and 6b, gradient descent only partially discovers suboptimal baseline candidates whereas the UNI algorithm discovers non-confounded and stable baselines for attribution.
>
> We again express our gratitude to the reviewer for their effort in improving our work and look forward to a fruitful discussion!

---

> > ### Comment · Reviewer_nTJj · 2024-11-26
> >
> > The authors have effectively addressed my concerns, leading me to increase my rating.

---

### Author Response · Authors · 2024-11-23
**General Response**

We thank the reviewers for their interest in our work and thoughtful feedback. We are encouraged that they found the ideas of UNI to be “innovative”, “novel” and “clearly explained”; they concur that UNI addresses an “important”, “previously overlooked” issue in interpretability that “[limits] static baseline approaches”; and that they recognised our experimental findings (of improved attribution faithfulness, robustness, stability) as being “well-structured”, “carefully designed [...] from different perspectives” and “methodically supported” with “careful mathematical analysis”. To further improve our work, we summarise major points of critique, and discuss how reviewers’ concerns are addressed and suggestions implemented. We further draw their attention to the revised manuscript, as well as responses to individual reviews.

**1 - Broader contributions to explainable AI (XAI).**
*1.1 - Granular unlearning for hierarchical feature/concept attribution.* UNI shows that first-order, sample-wise unlearning identifies and removes salient features predictive for a single sample. An exciting next step involves unlearning at the distribution level to localise and erase higher level concepts, such as unlearning a set of feature-clustered exemplars to erase common concepts. As noted by Reviewer Wrfy, unlearning in conjunction with activation matching can attribute features hierarchically, from local image textures, to intermediate shape prototypes, to class-semantic concepts.

*1.2 - Aligning XAI methods and models.* We have demonstrated how static baseline choices project assumptions onto the model, which result in undesirable biases that confound interpretation results. It follows to investigate how to design robust, unbiased XAI methods that do not contradict model behaviour. Model families of interest include equivariant models, models trained with spatial, geometric or colour augmentations, architectures with different inductive biases (ConvNets vs. transformers vs. state space models), and robust vs. non-robust models.

*1.3 - Comparing network inductive biases and interpretability during training.* With unbiased attribution methods, we can avoid introducing artefacts in attribution—UNI suggests that unlearning can remove confounders to visualise features that are responsible for the learning of a sample. As such, machine unlearning provides a strategy for rewinding the training process. This opens up possibilities of training-time interpretability and dissection of the intrinsic inductive biases native to different network architectures at initialisation.

**2. A theoretical motivating example for UNI.**
We strengthen theoretical arguments by analysing UNI under the simpler setting of a three gaussians mixture model, and study how our method preserves properties of monotonicity, low-curvature and proximity in 4.3 of the updated manuscript. We demonstrate that first-order unlearning removes salient features to discover a uniquely appropriate baseline. To understand the idea of activation matching, first consider that ReLU networks have decision boundaries that are well represented as piecewise linear functions. Activation matching supervises the unlearned baseline to use the same (piecewise linear) weights, which crucially satisfies the weak dependence property for conformal path-based attribution.

**3  - Scope and applicability: NLP extensions.**
We extend UNI experiments to the textual doimain, to interpret generative language models using activation patching. Activation patching is scalable but lacks faithfulness guarantees under causal interventions—UNI mitigates this by computing stable baselines. This enables interventional approaches to XAI and predictions of counterfactual model performance. Faithfulness results in Appendix Table 8 and Figure 9 speak to UNI’s general applicability to diverse data modalities, to both vision and language models, and to broader classes of interpretation methods.

---

### Meta-Review · Area_Chair_uFW3 · 2024-12-22

**Metareview:**

The authors study gradient-based, post-hoc, path attribution methods such as Integrated Gradients and identify that static baselines (e.g., black images, blurred noise) impose unintended biases on attribution maps, leading to fragility and unfaithful interpretations. They then proposed a novel framework which computes baselines by perturbing inputs in an "unlearning" direction (a direction the steepest ascent), to erase salient features while maintaining model-specific properties. The strong experimental results confirmed the intuition, demonstrating improved faithfulness, robustness, and stability.

The reviewers unanimously agree that it is a strong paper worthy of acceptance at ICLR.

**Additional Comments On Reviewer Discussion:**

During rebuttal, the authors further strengthened the paper by discussing broader contributions to explainable AI, extending experiments to the textual domain, and providing theoretical analysis under a simple setting of Guassian mixture models.

Three out of four reviewers raised their scores in response to the effective rebuttal, and the corresponding improvements made to the paper.

---

### Decision · Program_Chairs · 2025-01-22

Accept (Oral)